biochemistry/cellular biology/synthetic chemistry

Bub1, spindle assembly checkpoint, BAY-320, 2OH-BNPP1, kinase inhibitor

**Author for correspondence:**
Stephen S. Taylor
e-mail: stephen.taylor@manchester.ac.uk

# Inhibitors of the Bub1 spindle assembly checkpoint kinase: synthesis of BAY-320 and comparison with 2OH-BNPP1

Ilma Amalina[1], Ailsa Bennett[2], Helen Whalley[2], David Perera[2], Joanne C. McGrail[2], Anthony Tighe[2], David J. Procter[1] and Stephen S. Taylor[2]

[1]Department of Chemistry, School of Natural Sciences, University of Manchester, Oxford Road, Manchester M13 9PT, UK
[2]Division of Cancer Sciences, Faculty of Biology, Medicine and Health, University of Manchester, Manchester Cancer Research Centre, 555 Wilmslow Road, Manchester M20 4GJ, UK

IA, 0000-0001-9578-6680; HW, 0000-0002-2154-6999;
DP, 0000-0001-6678-8222; JCM, 0000-0002-7771-8526;
AT, 0000-0001-7173-232X; DJP, 0000-0003-3179-2509;
SST, 0000-0003-4621-9326

Bub1 is a serine/threonine kinase proposed to function centrally in mitotic chromosome alignment and the spindle assembly checkpoint (SAC); however, its role remains controversial. Although it is well documented that Bub1 phosphorylation of Histone 2A at T120 (H2ApT120) recruits Sgo1/2 to kinetochores, the requirement of its kinase activity for chromosome alignment and the SAC is debated. As small-molecule inhibitors are invaluable tools for investigating kinase function, we evaluated two potential Bub1 inhibitors: 2OH-BNPPI and BAY-320. After confirming that both inhibit Bub1 *in vitro*, we developed a cell-based assay for Bub1 inhibition. We overexpressed a fusion of Histone 2B and Bub1 kinase region, tethering it in proximity to H2A to generate a strong ectopic H2ApT120 signal along chromosome arms. Ectopic signal was effectively inhibited by BAY-320, but not 2OH-BNPP1 at concentrations tested. In addition, only BAY-320 was able to inhibit endogenous Bub1-mediated Sgo1 localization. Preliminary experiments using BAY-320 suggest a minor role for Bub1 kinase activity in chromosome alignment and the SAC; however, BAY-320 may exhibit off-target effects at the concentration required. Thus, 2OH-BNPP1 may not be an effective Bub1 inhibitor *in cellulo*, and while BAY-320 can inhibit Bub1 in cells, off-target effects highlight the need for improved Bub1 inhibitors.

# 1. Introduction

The spindle assembly checkpoint (SAC) is deployed by cells during mitosis to prevent segregation errors resulting from unattached or improperly attached chromosomes [1]. The SAC remains active at kinetochores until they have become stably attached to the spindle apparatus. Ultimately, SAC satisfaction leads to Cdc20 release from the inhibitory mitotic checkpoint complex allowing activation of the anaphase promoting complex, or cyclosome (APC/C) [2]. Once activated by Cdc20, the APC/C E3 ubiquitin ligase targets several proteins for degradation, including securin, ultimately leading to sister chromatin separation triggering anaphase onset.

SAC activity at the kinetochore is orchestrated by a network of protein interactions and the activity of several protein kinases, including Mps1, Aurora B and Bub1. Mps1 phosphorylation of MELT repeats of the Knl1 kinetochore protein enables SAC activation through recruitment of other SAC proteins such as Bub1 [3–7]. Aurora B is localized to centromeres via a combination of Haspin-mediated phosphorylation of histone H3 (H3pT3) and Bub1-mediated phosphorylation of histone H2A at Thr120 (H2ApT120), where it is required to promote correct kinetochore attachment and regulate the SAC [8].

Like Aurora B, Bub1 has been described as having a dual role in the SAC and chromosome alignment [9]. While its contribution to chromosome alignment has been consistently demonstrated [10–13], studies have yielded conflicting results regarding the requirement of Bub1 for SAC. Generation of conditional knockout mouse embryonic fibroblasts (MEFs) [10,14,15] and RNAi knockdown from HeLa and RPE1 cells [16,17] found Bub1 to be essential for SAC. Conversely, initial CRISPR-CAS9 genome editing approaches in RPE1 and HAP1 cells suggested only a minor role for Bub1 in the SAC when cells were sensitized via Mps1 inhibition [11,18]. These conflicting results were initially reconciled by the discovery that nonsense-associated alternative splicing allows for some Bub1 expression following CRISPR-CAS9 [19], and that siRNA knockdown of residual Bub1 greatly impaired SAC response in *BUB1*-disrupted cells [12]. However, more recently, HAP1 cells with several *BUB1* exons absent from genomic DNA were created following the use of two guide RNAs for CRISPR-CAS9 [20]. Surprisingly, the SAC remained functional in these cells, even when the more extensive approach was combined with Bub1 siRNA knockdown. However, the generation of a complete *BUB1* deletion was only possible in haploid HAP1 cells, but not in several other cell lines [11,20].

Despite the controversy, these experimental systems have allowed functional evaluation of Bub1. Bub1 has a Bub3-binding domain through which Bub1 is localized to kinetochores [21]. The Bub1 central region acts as a scaffold for Bub1-mediated localization of the RZZ complex, Mad1/2 and Cdc20 localization to kinetochores, and is required for the SAC function of Bub1 [11,12,15,17,22]. Numerous reports have found Bub1 kinase activity to be dispensable for SAC activation [11,13,14,17,22–24]; however, others suggest that Bub1-mediated phosphorylation of Cdc20 may directly contribute to APC/C inhibition [16,25,26]. Likewise, there are conflicting reports as to whether Bub1 kinase activity is required for chromosome alignment [11,14,17,23]. Bub1 H2ApT120 phosphorylation localizes Sgo1/2 and Aurora B, and other proteins of the chromosome passenger complex (CPC), to centromeres [11,13,14,17,23,27], and may integrate correction of attachment errors with SAC signalling [13]. In fact, H2ApT120 is required to maintain centromeric Aurora B and SAC activity in the absence of H3pT3 [28]. Finally, Bub1 is also autophosphorylated, both within the kinase domain activation segment and outside of this segment, which may have a role in regulating Bub1 localization [29,30].

Functional characterization of Bub1 would benefit from small-molecule kinase inhibitors. Indeed, drugs targeting Mps1 and Aurora B have been powerful tools for both deciphering kinase function and for dissection of mitotis [31–34]. A potent, specific Bub1 kinase inhibitor is of particular value since complete penetrance of genetic deletions or siRNA has been difficult in human cells, and only 4% of residual Bub1 is needed for SAC activity [12]. However, considering multiple conflicting reports regarding the function of Bub1, it is important that inhibitors used to evaluate its function are properly validated. The bulky ATP analogue 2OH-BNPP1 has been previously described as a Bub1 inhibitor and used to evaluate its function [16,35–38]; however, the *in cellulo* characteristics of this compound are not well reported. The substituted benzylpyrazole compounds, BAY-320 and BAY-524, were more recently shown to be highly selective inhibitors of Bub1 *in vitro* and to also inhibit its activity in cells [23,39]. Here, we compare 2OH-BNPP1 and BAY-320 using *in vitro* and cell-based assays to evaluate their relative merits as tools to probe the function of Bub1 kinase activity. Using these assays, we show that, while BAY-320 inhibited Bub1 both *in vitro* and in cells, 2OH-BNPP1 only inhibited Bub1 *in vitro* and did not effectively inhibit Bub1 in cells at the concentrations tested.

# 2. Results

## 2.1. Synthesis of BAY-320

To better define the role of Bub1 kinase activity in mitosis, a potent and selective Bub1 inhibitor would be of great value. We therefore sought to compare the characteristics of two previously described inhibitors, the bulky ATP analogue 2OH-BNPP1 [16] and the substituted benzylpyrazole compound BAY-320 [23,39]. We first sought to synthesize BAY-320 in our laboratory by adapting the methodology of Hitchcock *et al.* [39]. The eight-step synthesis, summarized in figure 1, started with the formation of Weinreb amide 2 from acyl chloride 1. Subsequent reaction of Weinreb amide 2 with ethylmagnesium bromide generated cyclopropyl ethyl ketone 3, which was then used to deliver 1,3-dicarbonyl 4. The pyrazole core in 5 was then formed using a Knorr reaction. Subsequent alkylation of the pyrazole core with benzyl bromide 6 gave ester 7 and functional group interconversion delivered amidine 8. After construction of the pyrimidine ring in 10, using reagent 9, cyclopropylbenzylpyrazole 11 (BAY-320) was obtained by Buchwald–Hartwig amination.

## 2.2. Both BAY-320 and 2OH-BNPP1 inhibit recombinant Bub1 *in vitro*

In order to confirm the ability of our synthesized BAY-320 to inhibit Bub1, and compare relative potency with 2OH-BNPP1, we first established and optimized an *in vitro* kinase assay using recombinant Bub1. First, HEK-293 cells were created that express tetracycline-inducible Bub1 with an N-terminal GFP tag to allow purification. A cell line expressing Bub1 with a lysine to arginine mutation in the catalytic motif (K821R) was also created as a catalytically inactive negative control [40]. When tetracycline-induced cells were exposed to nocodazole to maximize Bub1 activity [41,42], both wild-type and mutant Bub1 co-precipitated with Bub3 confirming expression of functional protein (electronic supplementary material, figure S1a) [21,43]. As expected, purified wild-type Bub1, but not K821R, was able to phosphorylate histone H2A (H2A-p) using $\lambda$-$^{32}$P-ATP *in vitro* (electronic supplementary material, figure S1b). Although the signal was weaker than for H2A-p, autophosphorylation of wild-type but not K821R Bub1 was also detected in this assay (Bub-1p). The *in vitro* kinase assay was then further optimized before evaluation of the small-molecule inhibitors to ensure linear velocity conditions (electronic supplementary material, figure S1c). Independently increasing either enzyme or substrate concentration amplified the H2A-p signal and quantification confirmed an initial linear relationship between both these parameters and the H2A-p product, which eventually plateaued at the higher enzyme concentrations evaluated. Subsequently, a Bub1 enzyme volume (10 µl beads) and H2A mass (2 µg) within the linear range were selected for further reactions. Unlike H2A-p, the Bub1 autophosphorylation signal remained unchanged up to the maximum H2A amount evaluated (4 µg). As expected, titration of unlabelled ATP from 50 to 400 µM into the reaction resulted in decreasing H2A-p signal generated from $\lambda$-$^{32}$P-ATP, and 100 µM unlabelled ATP was selected as optimal for subsequent assays. Conversely, titration of $\lambda$-$^{32}$P-ATP from 1–3 µCi per reaction, while maintaining 100 µM unlabelled ATP, gradually increased H2A-p signal with 2 µCi being selected as optimal (less than 0.2 µM). Finally, measurement of H2A-p production over time under final assay conditions found that maximum signal is obtained within 20 min. These parameters are similar to those used in previous Bub1 kinase assays [16,23].

Next, the optimized kinase assay was used to compare the ability of BAY-320 and 2OH-BNPP1 to inhibit Bub1 *in vitro*. Reactions did not include $\lambda$-$^{32}$P-ATP, rather immunoblotting for H2ApT120 was used measure kinase activity, which was evident with purified wild-type and not K821R Bub1 (figure 2a). Upon titration of both compounds into the assay, inhibition of Bub1 activity became apparent by 0.7 µM and the H2ApT120 signal was undetectable at 10 µM of inhibitor. These results confirmed that both inhibitors are able to inhibit Bub1 phosphorylation of H2A dose dependently *in vitro*. Quantification of immunoblots found the IC$_{50}$ of BAY-320 and 2OH-BNPP1 to be 0.56 and 0.60 µM, respectively (figure 2b). The results were consistent with the previous reports by Baron *et al.* for BAY-320 (IC$_{50}$ approximately 0.68 µM) [23], as well as by Kang *et al.* for 2OH-BNPP1 (IC$_{50}$ approximately 0.25 µM) [16]. In summary, both BAY-320 and 2OH-BNPP1 inhibited Bub1 kinase activity with similar potency by *in vitro* kinase assay.

As Bub1 kinase activity has previously been shown to be dispensable for the SAC [11,13,14,17,22–24], it was expected that the K821R mutant shown to be catalytically inactive *in vitro* would still be able to support a functional SAC. We decided to confirm this in the murine conditional *Bub1*-knockout system we previously created, using immortal MEFs harbouring tamoxifen-responsive Cre recombinase and a single *BUB1* allele floxed between two lox P sites (*BUB1$^{F/\Delta}$*) [10,14]. These MEFs could therefore be fully depleted of Bub1 protein using 4-hydroxy-tamoxifen (OHT). When treated with monastrol to prevent

**Figure 1.** Synthesis of BAY-320. Synthesis started with the formation of Weinreb amide 2 from acyl chloride 1. Subsequent reaction of Weinreb amide 2 with ethylmagnesium bromide generated cyclopropyl ethyl ketone 3, which was then used to deliver 1,3-dicarbonyl 4. The pyrazole core in 5 was then formed using a Knorr reaction. Subsequent alkylation of the pyrazole core with benzyl bromide 6 gave ester 7 and functional group interconversion delivered amidine 8. After the construction of the pyrimidine ring in 10, using reagent 9, cyclopropylbenzylpyrazole 11 (BAY-320) was obtained by Buchwald–Hartwig amination.

the formation of bipolar spindles, the MEFs activate SAC in the absence of OHT and can be seen to arrest in mitosis by time-lapse imaging [14]. However, in the presence of OHT to inactivate *Bub1*, monastrol treatment no longer results in a SAC response. When these cells are infected with recombinant adenoviruses expressing Bub1 cDNAs, both wild-type and Bub1 variants lacking either the whole kinase domain (ΔKD) or with a mutation in the catalytically important DFG motif (D919N; murine equivalent of D946N) are able to restore the SAC response in the presence of OHT [14]. Surprisingly, a Bub1 K795R mutant (murine equivalent of K821R) behaved similarly to a Δ38 Bub1 mutant protein lacking the Bub3-binding domain and was unable to restore a SAC response in the presence of OHT (electronic supplementary material, figure S2). This unexpected result emphasizes the need for a potent, highly selective inhibitor of Bub1 with activity in cells to allow for detailed analysis of its role in SAC. We therefore next developed a cell-based assay for Bub1 kinase activity.

## 2.3. BAY-320, but not 2OH-BNPP1, effectively inhibits Bub1 *in cellulo* at 10 µM concentration

Although BAY-320 has been previously shown to inhibit Bub1 *in cellulo* [23], the ability of the ATP analogue 2OH-BNPP1 to inhibit Bub1 in cells has not yet been unambiguously demonstrated. We therefore wanted to

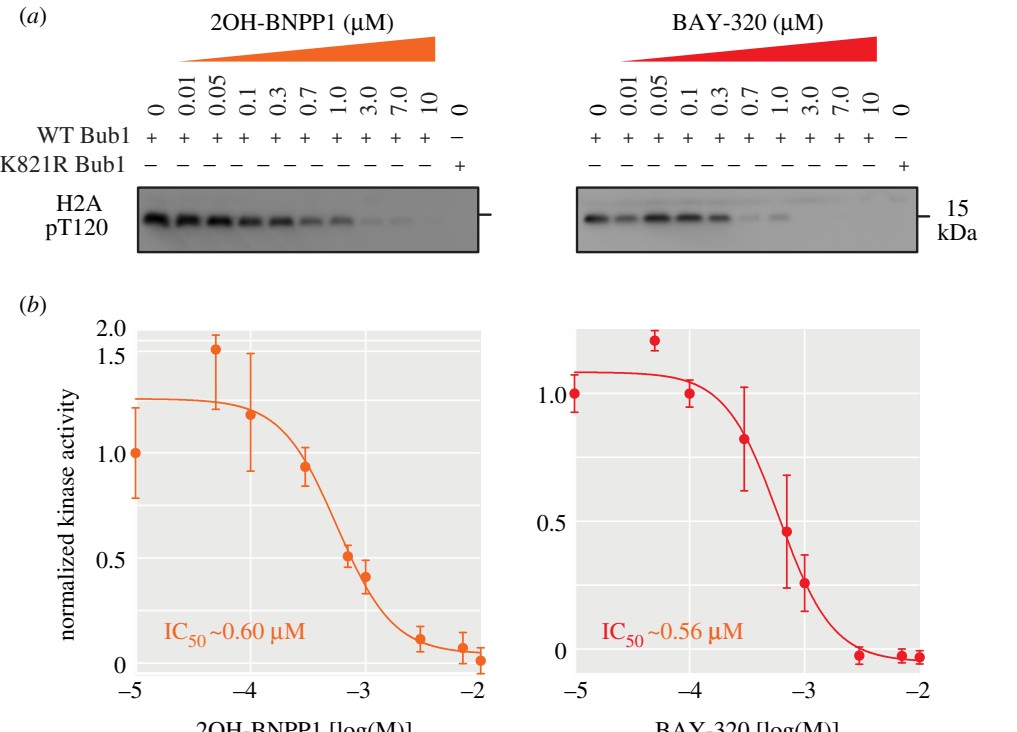

**Figure 2.** Both 2OH-BNPP1 and BAY-320 inhibit Bub1 kinase *in vitro*. (*a*) *In vitro* kinase assays using purified recombinant GFP-Bub1, H2A, and increasing doses of 2OH-BNPP1 and BAY-320 as indicated. Immunoblots for H2ApT120 production by wild-type Bub1 show dose-dependent inhibition of Bub1 kinase activity by 2OH-BNPP1 and BAY-320. The K821R GFP-Bub1 is also shown as a negative control. (*b*) Dose-response line graph from quantification of immunoblots normalized to one for maximum kinase activity, to calculate $IC_{50}$. Error bars show standard deviation from six independent replicates. Also see electronic supplementary material, figure S1.

develop a cell-based assay to directly compare the efficacy of the two compounds. To create an assay that was specifically dependent on Bub1 activity, we generated HeLa cells expressing a tetracycline-inducible GFP-tagged histone H2B protein fused to the kinase domain of Bub1 (Bub1C; residues 724–1085; figure 3*a*). Bub1C has been previously shown to be functionally similar to full-length Bub1 *in vitro* [16]. Such a fusion protein was used previously by S. Kawashima *et al.* [27] to specifically evaluate Bub1 kinase activity, as the H2B moiety tethers Bub1C to the chromosome arms where it can subsequently phosphorylate H2A allowing unambiguous visualization of Bub1C activity as ectopic H2ApT120 on the chromosome arms.

Tetracycline-inducible expression of the GFP-H2B-Bub1C fusion protein was first confirmed by immunoblotting (electronic supplementary material, figure S3a), then immunofluorescence was used to monitor H2ApT120 with and without its expression within cells arrested in mitosis by incubation with nocodazole for 16–18 h (figure 3*b*; electronic supplementary material, figure S3b). In the absence of tetracycline, immunofluorescence of H2ApT120 was detected at its expected centromeric location [27]. However, the expression of GFP-H2B-Bub1C in the presence of tetracycline caused H2ApT120 to re-locate along the chromosome arms in mitotic cells. In alignment with the role of H2ApT120 in Sgo1 recruitment to centromeres, delocalization of H2ApT120 also resulted in Sgo1 to be located on the arms (electronic supplementary material, figure S3b) [14,27]. Furthermore, this distribution of H2ApT120 and Sgo1 on the chromosome arms was not seen in the presence of a GFP-H2B-Bub1C fusion protein harbouring the D946N mutation of the human DFG motif of the kinase domain (electronic supplementary material, figure S3b). These observations confirm that the transgene was behaving as anticipated. Interestingly, expressing GFP-H2B-Bub1C, but not the D946N mutant, induced H2ApT120 throughout the nucleus in interphase cells (electronic supplementary material, figure S3c), thereby facilitating quantitation of large numbers of cells (see below). Nevertheless, these observations show that inducing GFP-H2B-Bub1C results in ectopic phosphorylation of H2A on chromosome arms in mitosis, providing a cell-based assay to analyse Bub1 kinase activity.

Cells expressing the wild-type GFP-H2B-Bub1C fusion protein were then used to test the ability of 2OH-BNPP1 and BAY-320 to inhibit Bub1C. Exposure to 10 µM BAY-320 for 3 h prevented the distribution of H2ApT120 along chromosome arms in tetracycline-induced cells; however, H2ApT120 remained visible on the arms in the presence of 10 µM 2OH-BNPP1 (figure 3*b* highlighted panels).

(*a*)

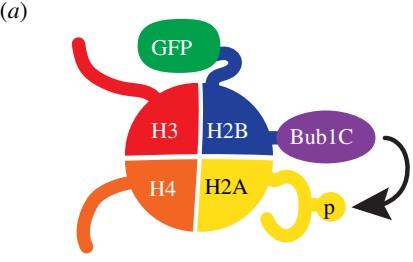

(*b*)

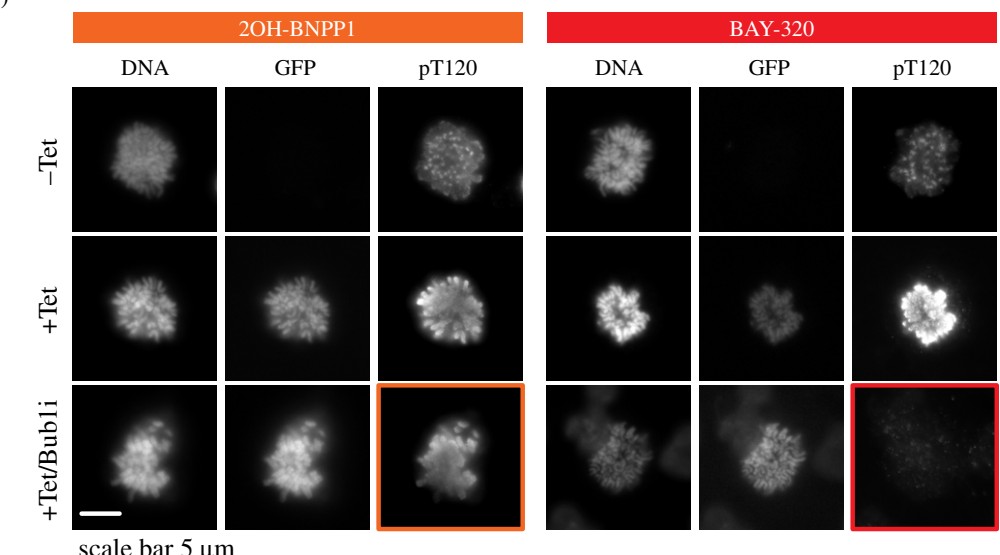

scale bar 5 μm

(*c*)

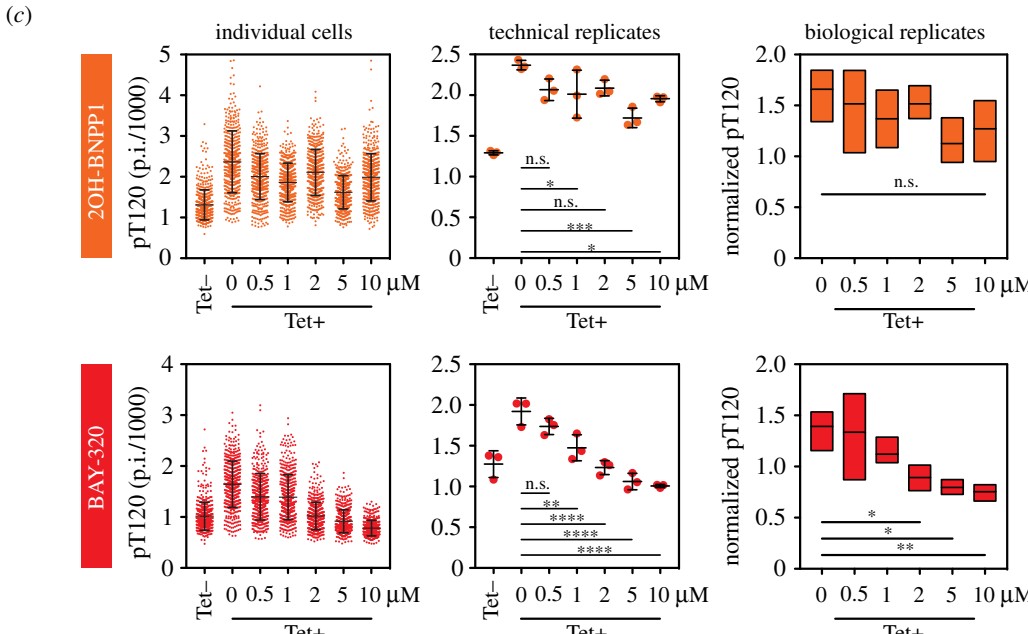

**Figure 3.** (*Caption opposite.*)

This suggested that the Bub1C fusion remained catalytically active in the presence of 2OH-BNPP1 but was being inhibited by BAY-320. Analysing large numbers of cells upon titration of the inhibitors and quantification of staining intensity from individual cells showed a progressive decrease in H2ApT120 intensity as the concentration of BAY-320, but not 2OH-BNPP1, was increased (figure 3*c*). In fact, compared with the absence of inhibitor, a significant reduction in H2ApT120 staining was consistently seen at BAY-320 concentrations of 2 μM or higher with both technical and biological replicates. However, such significant reductions in H2ApT120 staining were not consistently seen with increasing

**Figure 3.** (*Opposite.*) BAY-320 inhibits ectopically expressed Bub1 kinase in cells. (*a*) Schematic of histone octamer incorporating the GFP-H2B-Bub1C fusion protein, which subsequently tethers Bub1C to chromatin where it can phosphorylate H2A. Bub1C includes Bub1 residues 724–1085, including the serine/threonine kinase domain and N-terminal extension required for kinase activity [16]. (*b*) Immunofluorescence images of nocodazole-treated mitotic HeLa cells following tet-induction for 16–18 h and staining for DNA and H2ApT120 (pT120). In the bottom images, cells expressing GFP-H2B-Bub1C were also treated with 10 μM 2OH-BNPP1 (+Tet/Bub1i; left images) or 10 μM BAY-320 (+Tet/Bub1i; right images) for 3 h prior to fixing and staining. Highlighted images show that de localized staining of pT120 resulting from expression of GFP-H2B-Bub1C is effectively inhibited by 10 μM BAY-320, but is not effectively inhibited by 10 μM 2OH-BNPP1. Scale bar, 5 μm. (*c*) pT120 immunofluorescence quantification from HeLa cells, with or without 16–18 h tetracycline treatment, and with exposure to 2OH-BNPP1 (upper panels) or BAY-320 (lower panels) for 3 h at concentrations indicated; (left panels) shows immunofluorescence quantification (fluorescence pixel intensities) from 500 cells per condition from a single technical replicate in one experiment; (centre panels) shows a dot plot with the means of three technical replicates from one experiment. Lines in (left panels) and (centre panels) show the mean ± s.d. In (right panels), boxes show the median and interquartile ranges from three independent biological replicates, each based on three technical replicates analysing 500 cells each. Ordinary one-way ANOVA with Dunnett's multiple comparisons test. n.s., not significant; $^{****}p < 0.0001$; $^{***}p < 0.001$; $^{**}p < 0.01$; $^{*}p < 0.05$. Tet, tetracycline. Also see electronic supplementary material, figure S3.

concentrations of 2OH-BNPP1. Therefore, although both compounds are able to inhibit Bub1 *in vitro*, we found that only BAY-320 is an effective inhibitor of cellular Bub1 and inhibition by 10 μM 2OH-BNPP1 was not apparent in our Bub1-specific cell-based assay.

## 2.4. BAY-320 inhibits endogenous Bub1 and reduces centromeric localization of Sgo1

Having demonstrated Bub1 inhibition by BAY-320 *in cellulo* using overexpressed exogenous enzyme, we wanted to confirm our findings by evaluating the impact of BAY-320 and 2OH-BNPP1 on endogenous Bub1. In the absence of inhibitor, both H2ApT120 and Sgo1 can be seen to localize to centromeres in HeLa cells arrested in mitosis using nocodazole (figure 4*a,b*). Similarly, with the addition of 10 μM 2OH-BNPP1, strong staining of both H2ApT120 and Sgo1 at centromeres was still apparent in mitotic cells. By contrast, the addition of 10 μM BAY-320 for 3 h resulted in delocalization of Sgo1 (figure 4*a*) and almost completely abolished centromeric staining of H2ApT120 (figure 4*b*). These results support findings from the GFP-H2B-Bub1C fusion protein on endogenous substrates; while BAY-320 is able to inhibit the kinase activity of Bub1 both *in vitro* and *in cellulo*, 2OH-BNPP1 appears to only effectively function as an inhibitor of Bub1 kinase *in vitro* and does not inhibit Bub1 in cells at 10 μM concentration.

## 2.5. BAY-320 treatment results in aberrant mitoses

Previously, Baron *et al.* [23] found that the inhibition of Bub1 with BAY-320 had only a minimal impact on mitotic progression in HeLa cells, with no impact on untransformed RPE1 cells. We decided to further evaluate the consequences of Bub1 inhibition on mitosis using the DLD-1 colon cancer cell line, which maintains a 'flatter' morphology during mitosis allowing easy visualization of chromosomes. A DLD-1 cell line expressing GFP-H2B was used to allow monitoring of chromosome movement within asynchronous growing cells using fluorescence time-lapse imaging (figure 5*a*) [44]. In the absence of inhibitor, DLD-1 cells took on average approximately 83 min to complete mitosis. Compared with control cells, treatment with 10 μM BAY-320 significantly prolonged the time required for the DLD-1 cells to complete mitosis, with cells treated with inhibitor spending approximately 117 min in mitosis (figure 5*a,b*). This prolonged duration of mitosis with BAY-320 treatment appeared to be the result of additional time taken to align the chromosomes before division compared with the control cells (figure 5*a*). Even following BAY-320 treatment, the majority of cells completed mitosis normally; however, we did observe a significant increase in the number of aberrant mitoses with BAY-320 treatment versus control cells (figure 5*b*; 24% of cells treated with BAY-320 versus 13% of control cells). The predominant aberrant mitotic characteristics resulting from BAY-320 treatment were anaphase bridges (34%), anaphase with unaligned chromosomes (27%) and micronuclei (30%). Taken together, these results suggest that while not essential for chromosome alignment, Bub1 kinase assay activity may contribute to the efficient alignment of chromosomes.

## 2.6. Treatment with BAY-320 impacts cell survival in colony-forming assays

As Bub1 inhibition with BAY-320 appears to impact mitosis, albeit modestly, next we decided to evaluate the impact of BAY-320 on cell survival using a colony-forming assay. Since Bub1 inhibitors are under

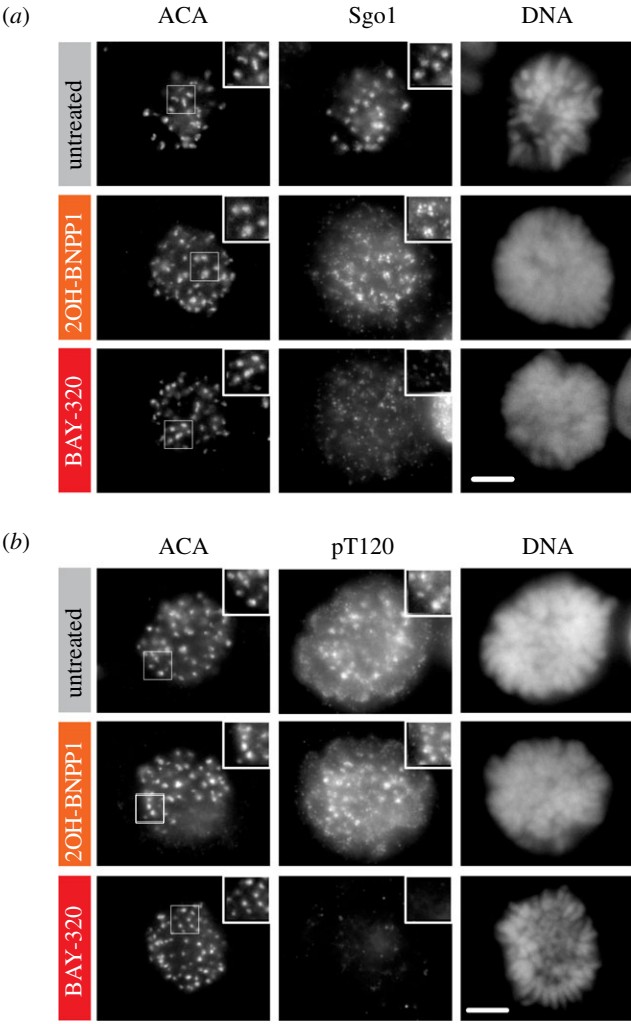

**Figure 4.** BAY-320 inhibits centromeric localization of Sgo1. Immunofluorescence images of the mitotic nuclei of HeLa cells following treatment with nocodazole for 16–18 h without inhibitor (untreated, top images), or treated with 10 µM 2OH-BNPP1 (centre images) or 10 µM BAY-320 (bottom images) for 3 h prior to fixing and staining. (a) Immunofluorescence showing loss of centromeric Sgo1 in the presence of 10 µM BAY-320, but not 10 µM 2OH-BNPP1. (b) Immunofluorescence showing loss of centromeric H2ApT120 (pT120) in the presence of 10 µM BAY-320, but not 10 µM 2OH-BNPP1. Scale bar for (a) and (b) 5 µm.

evaluation for their anti-cancer potential [24], for this assay, we compared three different human cell lines: the ovarian cancer cell lines OVCAR-3 and Kuramochi, and the non-transformed cell line RPE1 for comparison. RPE1 cells were also used for consistency with previous analyses of BAY-320 [23]. As 2OH-BNPP1 is not able to effectively inhibit Bub1 in cells, we used it as a negative control in these assays. Cell lines were treated with BAY-320 or 2OH-BNPP1 for 3 days followed by wash out and then stained after 6–19 days to visualize colony formation. For all cell lines, 5 µM of either BAY-320 or 2OH-BNPP1 had no impact on the ability of cells to form colonies (figure 6a). However, treatment with BAY-320 at 10 µM resulted in a substantial reduction in colony formation, particularly of Kuramochi cells. As expected, even 10 µM 2OH-BNPP1 treatment had no impact on the ability of any of the cell lines to form colonies. These results are in agreement with the cell-based assays for Bub1 kinase assay activity demonstrating that 2OH-BNPP1 does not inhibit Bub1 in cells at concentrations of up to 10 µM.

The strong inhibition of colony formation following washout of BAY-320 of the two ovarian cancer cell lines with p53 mutations [45,46] prompted us evaluate the impact of p53 deletion on cell survival following treatment with BAY-320. It has been previously suggested that p53 has a role in monitoring Bub1 function, and cells deficient in both Bub1 and active p53 are highly aneuploid [47]. We therefore repeated our colony-forming assay using the *TP53* wild-type RKO1 cell line, from which we engineered *TP53*$^{-/-}$ RKO1 cells using CRISPR-CAS9 disruption, allowing direct comparison of isogenic cells with and without p53. While wild-type RKO1 cells expressed p53, which could be

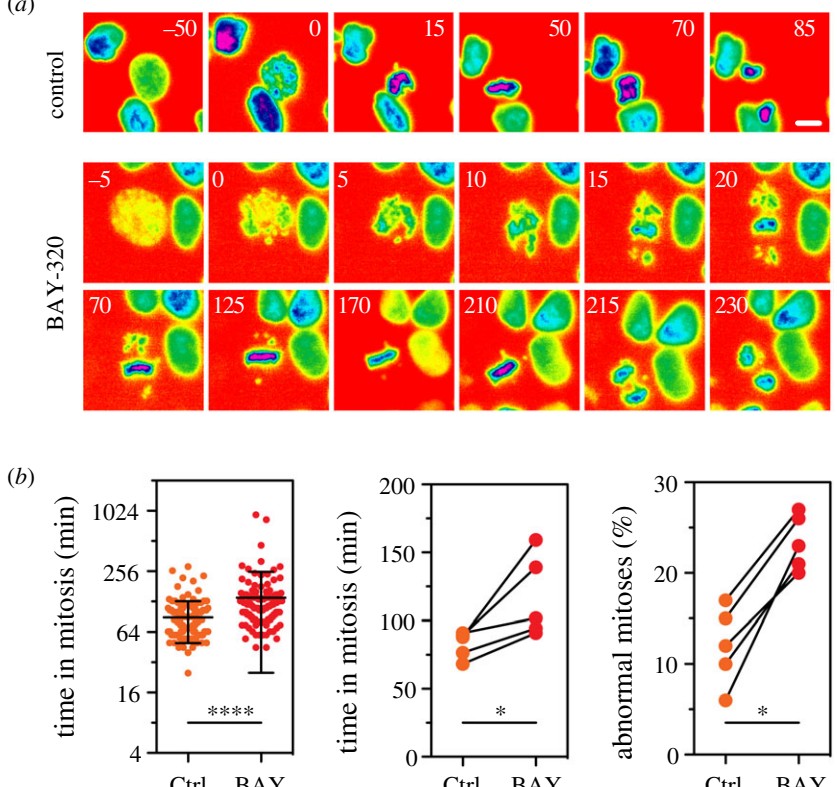

**Figure 5.** Cells treated with BAY-320 exhibit chromosome mis-segregation. (*a*) Time-lapse image sequences of asynchronous DLD-1 cells expressing GFP-H2B, showing prolonged time to complete mitosis following treatment for 3 days with 10 µM BAY-320 (lower images), compared with control cells treated with DMSO (upper images). Numbers show minutes after imaging initiated. Images were taken every 5 min for 72 h. Scale bar, 10 µm. (*b*) Time in mitosis and proportion of abnormal mitosis following treatment for 3 days with DMSO (Ctrl) or 10 µM BAY-320 (BAY) as determined by manual fluorescent time-lapse analysis. The time in mitosis is defined as the interval between nuclear envelope breakdown and chromosome de-condensation. Scatter plot in (left panel) is the time spent in mitosis by individual DLD-1 cells under each condition from a single experiment (*n* = 139 cells for control; *n* = 124 cells for BAY-320). Lines show the mean ± s.d. Mann–Whitney test, ****$p$ < 0.0001. Ladder plot in (centre panel) shows mean time in mitosis from five independent experiments. Ladder plot in (right panel) shows the mean proportion of abnormal mitoses from five independent experiments. For (centre and right panels), a total of 1201 control and 1408 drug-treated cells were analysed across the five independent experiments. Paired *t*-test; *$p$ < 0.05.

further induced in the presence of the Mdm2 inhibitor Nutlin-3, the $TP53^{-/-}$ RKO1 cell line did not express p53 in the presence or absence of Nutlin-3 (figure 6*b*). Subsequently, these cells were treated with increasing concentrations of the inhibitors for 3 days before washout and plating to allow colony formation. Again, 2OH-BNPP1 did not impact the ability of either RKO1 cell line to form colonies after washout of the inhibitor (figure 6*c*). In alignment with RPE1 and ovarian cancer cell lines, BAY-320 inhibited colony formation of both the wild-type and $TP53^{-/-}$ RKO1 cells at 10 µM concentration. The level of inhibition was similar in both RKO1 cell lines, even at 12.5 µM BAY-320, therefore, the loss of p53 function does not appear to sensitize cells to Bub1 inhibition. However, once again, these results confirm the activity of BAY-320 in multiple different human transformed and non-transformed cell lines.

## 2.7. Cell fate profiling reveals cancer cell death with BAY-320 treatment

To help better understand how Bub1 inhibition impacts mitosis and cell fate, we set out to record the individual cell fate profiles of OVCAR-3 and Kuramochi, and RPE1 cells in the presence and absence of BAY-320 over 3 days. In the absence of drug treatment, the majority of RPE1 cells (80%) underwent normal mitosis, dividing two or three times over the 3 days (figure 7). The remaining 20% of cells did not enter mitosis. The untreated profiles of the cancer cell lines differed from the untransformed RPE1 cells in that a number of cells died in either interphase or mitosis during the profiling. However, around half of the

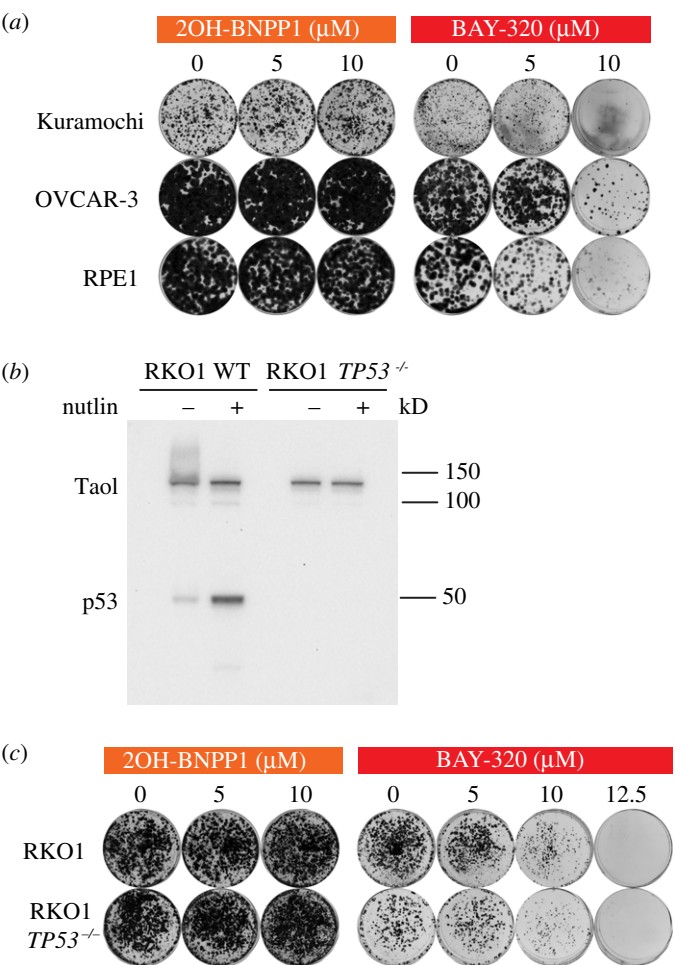

**Figure 6.** BAY-320 treatment impacts the survival of various cell lines. (*a*) Colony formation of OVCAR-3 and Kuramochi cell lines, and the non-transformed cell line RPE1, treated with indicated concentrations of 2OH-BNPP1 or BAY-320 for 3 days before washout. Colony staining was 6 days (RPE1), 19 days (Kuramochi) or 13 days (OVCAR-3) after washout. (*b*) Immunoblot of wild-type and $TP53^{-/-}$ RKO1 cells for p53 in the presence and absence of the Mdm2 inhibitor Nutlin-3, which stabilizes p53. (*c*) Colony formation from RKO1 and RKO1 $TP53^{-/-}$ cells, treated with indicated concentrations of 2OH-BNPP1 or BAY-320 for 3 days before washout and subsequent colony staining after 6 days.

OVCAR-3 and the majority of the Kuramochi cells still underwent normal mitoses, also undergoing two or three divisions. Next, we recorded the cell fate profiles over 3 days while cells were exposed to 10 µM BAY-320. The fates experienced by all three cell lines in the presence of BAY-320 appeared to differ from their corresponding untreated profiles; however, there were also differences in how the RPE1 cells and the cancer cells responded to drug treatment (figure 7). Treatment of the RPE1 cells resulted in a reduction of proliferation, with the proportion of cells not entering mitosis increasing from 20% without treatment to 37% with BAY-320 treatment. In addition, the number of divisions completed by RPE1 within the 3-day period also appeared to be reduced, compared with the untreated RPE1 cells, with most dividing cells only undergoing one division. By contrast, treatment of both cancer cell lines with BAY-320 increased cell death compared with the corresponding untreated profiles. Death of the Kuramochi cells increased from 16% of the untreated cells to 64% of the treated cells. Surprisingly 57% of the treated cells died during interphase without entering mitosis. Treatment resulted in the death of 100% of the OVCAR-3 cells by the end of 3 days, again with the majority of cells dying during interphase (84%). Taken together, these data corroborate previous observations and show that treatment with BAY-320 certainly has an impact on the fate of cells, compared with the corresponding untreated cells, resulting in fewer divisions in the case of RPE1 and increased cell death in the case of OVCAR-3 and Kuramochi cells. However, the extent of death during interphase without a preceding mitosis is perplexing. Consequently, these observations need to be interpreted with caution, e.g. it is possible that death in these cell lines is due to an off-target drug effect and/or a consequence of the experimental conditions.

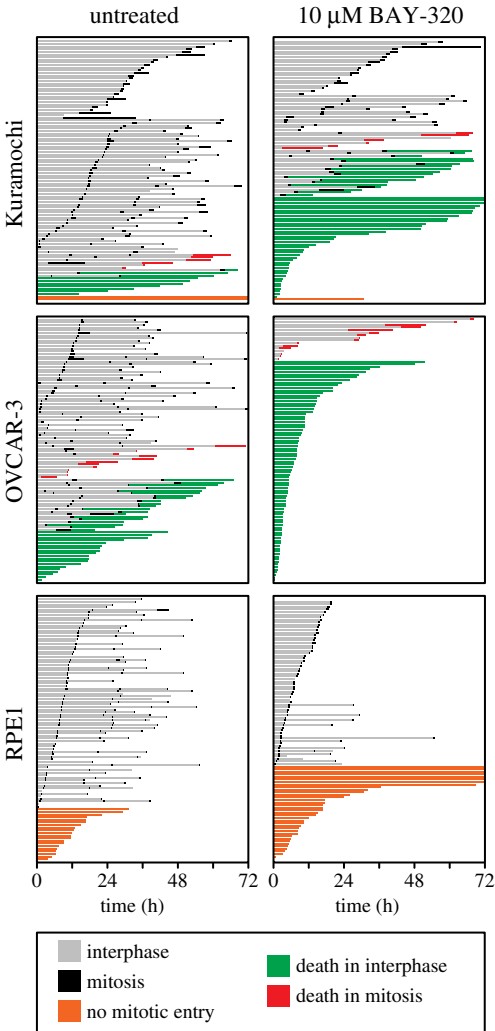

**Figure 7.** Cell fate profiling reveals cancer cell death with BAY-320 treatment. Cell fate profiles over 72 h determined by time-lapse microscopy of OVCAR-3 and Kuramochi and RPE1 cells. Cells were untreated (left panels) or treated with 10 µM BAY-329 (right panels) immediately prior to time-lapse starting at T0, with images acquired every 10 min. Each horizontal bar represents a single cell with the colours indicating cell behaviour. One hundred randomly selected cells were analysed per condition in a single experiment.

## 3. Discussion

Development and comprehensive characterization of small molecules targeting mitotic regulators is important not only due to their anti-cancer potential, but also as they are invaluable tools for deciphering the intricate workings of mitotic processes [31–34]. A potent, specific Bub1 kinase inhibitor is of particular value since Bub1 has a complex role in both chromosome alignment and SAC signalling [9], with studies yielding conflicting results regarding the requirement of Bub1 for the SAC [20]. In particular, the role of Bub1 kinase activity in SAC signalling remains controversial, with some reports showing importance [16,25,26], and others indicating that Bub1 kinase activity is redundant for SAC activity [11,13,14,17,22–24]. Therefore Bub1-specific inhibitors represent a useful tool for clarification of its role. Here, we characterized two previously identified Bub1 inhibitors, 2OH-BNPP1 and BAY-320 [16,23,39]. Although both these inhibitors have been used to evaluate Bub1 function [35,36,38], our data suggest that BAY-320 represents a better tool for cell-based studies.

We first confirmed that both 2OH-BNPP1 and BAY-320 were able to effectively inhibit H2ApT120 phosphorylation *in vitro*, and the IC$_{50}$s determined here, 0.56 µM for BAY-320 and 0.60 µM for 2OH-BNPP1, are in agreement with previous reports [16,23]. Therefore, we set out to compare the agents in cells using an assay based on the fusion protein described by Kawashima *et al.* [27], which artificially provides a direct outcome of Bub1 kinase activity to eliminate interference from off-target effects. We fused Bub1C, including the kinase domain and N-terminal extension required for kinase activity [16],

to H2B to ectopically delocalize Bub1 kinase activity from centromeres by placing it in close proximity to its H2A substrate within chromatin universally along chromosome ams. Overexpression of the fusion protein was tetracycline-inducible meaning that Bub1C activity could be easily detected as ectopic H2ApT120 along the chromosome arms in the presence of tetracycline. Using this assay, BAY-320 effectively inhibited the ectopic H2ApT120 signal; however, this Bub1C-specific H2ApT120 signal remained with 2OH-BNPP1 treatment at the range of concentrations tested.

Previously, Liu *et al.* [35] found that 2OH-BNPP1 treatment of mitotic HeLa cells reduced H2ApT120 localization to centromeres and subsequently inhibited centromeric localization of Sgo1. However, we were unable to demonstrate inhibition of either overexpressed H2B-Bub1C or endogenous Bub1 by 2OH-BNPP1 in HeLa cells. In fact, while Nyati *et al.* using 2OH-BNPP1 proposed that SMAD2/3 proteins were Bub1 substrates upon TGFβ stimulation [38], it was subsequently found that only a TGFβ inhibitor and not BAY-320, or the novel Bub1 inhibitor BAY-1816032, was able to block TGFβ-dependent SMAD2/3 phosphorylation [24]. These authors found 2OH-BNPP1 to be unselective for Bub1 and able to inhibit multiple other kinases at $IC_{50}$s of 30–100 nmol l$^{-1}$. This unselectively may explain why 2OH-BNPP1 is an effective Bub1 inhibitor *in vitro* but does not effectively inhibit Bub1 in cells. Following Lipinski's rule, given the 2OH-BNPP1 molecular weight of 297 g mol$^{-1}$ and the lipophilicity logP value of 3.01 (by ChemDraw prediction), it is unlikely that cell permeability is limiting 2OH-BNPP1 activity as this value is in accordance with the 'rule of 5' for which compounds are likely to be cell permeable [48,49]. Nonetheless, our H2B-Bub1C assay provides a robust tool for validating Bub1 inhibitors in cells.

In line with results from Baron *et al.* [23], we find that BAY-320 synthesized following our protocol is able to inhibit Bub1 both *in vitro* and *in cellulo*. In both studies, BAY-320 treatment resulted in a reduction in centromeric levels of Sgo1 resulting from the absence of H2ApT120 at centromeres. Here we found, following time-lapse imaging of DLD-1 colon cancer cells, that Bub1 inhibition with 10 µM BAY-320 resulted in an albeit minor but significantly increased time to complete mitosis compared with untreated cells. This prolonged time in mitosis appeared to be due to inefficient chromosome alignment, suggesting that Bub1 kinase activity has a role in chromosome alignment in these cells, as seen previously with HeLa cells [17]. In addition, compared with controls, a significantly greater proportion of DLD-1 cells underwent aberrant mitoses with BAY-320 treatment, also suggesting a minor defect in SAC signalling may be allowing cells with unaligned chromosomes to undergo anaphase. In contrast with our data, Baron *et al.* [23] did not see a chromosome alignment defect in HeLa cells treated with 10 µM BAY-320. In addition, in their experiments, Bub1 inhibition had only a very subtle impact on the duration of mitosis, which was only evident in HeLa cells and not non-transformed RPE1 cells, although treatment was only at 3 µM BAY-320. Surprisingly, differences between cell lines observed were also apparent upon Bub1 depletion, whereby Bub1-depleted HeLa cells experienced significantly prolonged mitoses; however, Bub1-depleted RPE1 cells were unaffected versus control treatment [23]. Taken together with our observations, dependency on Bub1 protein and its kinase activity may vary across cell lines, as seen with CRISPR-CAS9 disruption of *BUB1* [20]. It may therefore be of interest to compare the impact of Bub1 inhibition on additional cell lines to see if more sensitive lines can be identified. Baron *et al.* [23] suggested that, although not absolutely required for SAC, Bub1 kinase activity has a minor role in the maintenance of the SAC signal. A greater reliance of DLD-1 cells, versus HeLa or RPE1, on Bub1 kinase activity may explain the differential results we see here.

Although not examined here, Baron *et al.* [23] also studied localization of CPC components following Bub1 inhibition. Following treatment with 3 µM BAY-320 or Bub1 depletion, CPC components were partially displaced from centromeres, with displacement being greater in the Bub1-depleted versus inhibited HeLa cells. This residual CPC at centromeres was suggested to support chromosome alignment when Bub1 is inhibited [23]. Subsequently, combination treatment with BAY-320, or the novel Bub1 inhibitor BAY-1816031, with paclitaxel resulted in severe chromosome segregation defects [23,24]. BAY-1816031 plus paclitaxel combined also increased the duration of mitosis in HeLa cells compared with controls [24]. Therefore, it was proposed that in the presence of both paclitaxel and Bub1 inhibitors, residual CPC at centromeres may not be sufficient to support effective alignment [23]. In our experiments, the increased dose of BAY-320 may result in greater displacement of CPC components from centromeres, than lower doses of BAY-320, thereby prolonging time in mitosis. Likewise, in addition to dose dependency, the levels of residual CPC at centromeres following Bub1 inhibition may vary across cell lines.

We also evaluated the impact of prolonged treatment with 10 µM BAY-320 on cell proliferation and survival. Colony-forming assays showed that BAY-320 impaired growth of all the cell lines that we

tested, including cancer cell lines and RPE1 cells. Baron *et al.* [23] did not see an impact of BAY-320 monotherapy on colony formation; however, again they exposed cells to a lower concentration of BAY-320 (3 µM). As our results suggested toxicity of BAY-320 at 10 µM, we also looked at the impact of 10 µM BAY-320 on cell fate and again saw differences between cell lines. While treatment of the RPE1 cells resulted in a reduction of proliferation, the treatment of both ovarian cancer cell lines increased cell death compared with corresponding untreated profiles. However unexpectedly, given the role of Bub1 in mitosis, cell death resulting from treatment mainly occurred during interphase. Constitutive low-level activation of Bub1 in interphase has been proposed previously; however, this is autophosphorylation not H2ApT120 [29,30]. We did find that overexpressed Bub1C demonstrated H2ApT120 activity during interphase, presumably because it lacks a regulatory element or it is improperly regulated when overexpressed. However, non-mitotic functions of Bub1 have been proposed, for example in the DNA damage response [50,51] or telomere replication [37], which could be responsible for the interphase phenotype observed here. Nevertheless, it is possible that the cell fate profiles of the ovarian cancer cell lines are due to off-target effects of BAY-320 during interphase. Although Baron *et al.* [23] found 10 µM BAY-320 to have modest activity against other kinases *in vitro*, they only confirmed lack of inhibition of the functionally relevant Haspin in cells when other more prominent off-target kinases were identified *in vitro*. Comparison of the impact of treatment on Bub1 wild-type and Bub1-knockout cells will utimately be required to confirm the presence or absence of off-target effects. Nonetheless, interpretation of our results is somewhat limited, given the high proportion of interphase deaths of the cancer cell lines in the absence of BAY-320 treatment, and further study is warranted. Although BAY-320 has potentially provided some insight into Bub1 function, the high concentration required and the potential off-target effects highlight the ongoing need for superior Bub1 inhibitors. It was also recently reported that the pharmacodynamic properties of BAY-320 have been found to be limiting for animal studies [24].

An evaluation of the novel Bub1 inhibitor BAY-1816031 has now been reported [24]. The specificity of this agent was first interrogated against multiple kinases *in vitro* and then activity against its most prominent off-target kinase (LOK) confirmed to be absent in cells, supporting Bub1 selectivity [24]. Again, it was found that BAY-1816031 had minimal impact on the SAC, although it has been reported to inhibit Bub1-mediated H2ApT120 and subsequent Sgo1 centromere localization [24,52]. BAY-1816031 monotherapy was found to inhibit the proliferation of multiple cancer cell lines at low concentrations ($IC_{50}$ 0.5–5.8 µmol l$^{-1}$); however, unfortunately this screen did not identify highly resistant or sensitive cell lines [24]. As well as paclitaxel, BAY-1816031 exhibits promising anti-proliferative activity against cancer cell lines when combined with ATR or PARP inhibitors and significantly reduces tumour size in breast cancer xenografts when combined with the PARP inhibitor olaparib [24]. As BAY-1816031 also exhibits a favourable pharmacokinetic and safety profile [24], future evaluation in clinical studies is anticipated. BAY-1816031 is also expected to be of value for further study of Bub1 function, although our results highlight the need for vigorous examination of inhibitors before conclusions are drawn based on their impact on cells.

# 4. Experimental Procedures

## 4.1. Synthesis of BAY-320

The synthesis of compounds 2–11 (figure 1) followed procedures by Hitchcock *et al.* [39] with some modification of methods. Figures generated using ChemDraw 18.0 (PerkinElmer).

## 4.2. *N*-Methoxy-*N*-methylcyclopropanecarboxamide[x]

*N,O*-Dimethylhydroxylamine hydrochloride (8.26 g, 84.7 mmol, 1.1 eq.) and triethylamine (23 ml, 169 mmol, 2.2 eq.) were dissolved in 160 ml of dichloromethane and the solution cooled to 0°C. Cyclopropanecarbonyl chloride **1** (7.00 ml, 77.0 mmol, 1.0 eq.) was added dropwise at 0°C and then allowed to reach room temperature and stirred for 18 h before washing with water, saturated NaHCO$_3$, 1 M HCl and brine. The organic layers were combined, dried over MgSO$_4$ and concentrated

under vacuum to obtain the title product as a yellow liquid (8.22 g, 54.0 mmol, 83%). $^1$H NMR (500 MHz, CDCl$_3$) $\delta$ ppm 0.77 (dd, 2H, $J$ = 7.0, 3.7 Hz, CH$_2$), 0.93 (dd, 2H, $J$ = 6.8, 3.7 Hz, CH$_2$), 2.10 (apparent s broad, 1H, CH), 3.17 (s, 3H, CH$_3$), 3.72 (s, 3H, CH$_3$); $^{13}$C NMR (101 MHz, CDCl$_3$) $\delta$ ppm 7.9 (2x CH$_2$), 9.9 (CH), 32.6 (NCH$_3$), 61.6 (OCH$_3$), 174.9 (C(O)); IR $\nu_{max}$ (thin film, cm$^{-1}$) = 3005, 2966, 1651, 1474, 1178, 1001, 754; ESI MS $m/z$ for C$_6$H$_{11}$NO$_2$ [M + Na]$^+$: 152.0.

## 4.3. 1-Cyclopropylpropan-1-one[x]

Ethylmagnesium bromide solution in 3M diethyl ether (15.5 ml, 46.5 mmol, 1.2 eq.) was added to diethyl ether (70 ml) and cooled to −78°C. N-methoxy-N-methylcyclopropanecarboxamide **2** (4.22 ml, 38.8 mmol, 1.0 eq.) in diethyl ether (8 ml) was added dropwise and the reaction mixture stirred for 18 h while warming to room temperature before quenching with ammonium chloride (25 ml) and water (25 ml). The organic layers were extracted with diethyl ether, washed with brine, dried over MgSO$_4$, filtered and concentrated to obtain the crude product as a yellow liquid. Product **3** was then used in the next step without further purification. $^1$H NMR (500 MHz, CDCl$_3$) $\delta$ ppm 0.85 (dd, 2H, $J$ = 7.2, 3.6 Hz, CH$_2$), 1.01 (d, 2H, $J$ = 3.8 Hz, CH$_2$), 1.09 (t, 3H, $J$ = 7.3 Hz, CH$_3$), 1.92 (ddd, 1H, $J$ = 12.4, 7.9, 4.6 Hz, CH), 2.58 (q, 2H, $J$ = 7.3 Hz, CH$_3$CH$_2$); $^{13}$C NMR (101 MHz, CDCl$_3$) $\delta$ ppm 7.7 (CH$_3$), 10.3 (2x CH$_2$), 19.8 (CH), 36.33 (C(O)CH$_2$CH$_3$), 211.4 (C(O)); IR $\nu_{max}$ (thin film, cm$^{-1}$) = 3018, 1696, 1386, 1216, 753; ESI MS $m/z$ calculated for C$_6$H$_{10}$O [M + H]$^+$: 99.0.

## 4.4. 4-Cyclopropyl-3-methyl-2,4-dioxobutanoate[x]

LiHMDS in 1M THF (20.4 ml, 20.4 mmol, 2.0 eq.) was added to diethyl ether (95 ml) and the solution cooled to −78°C. 1-Cyclopropylpropan-1-one **3** (1.00 g, 10.2 mmol, 1.0 eq.) in diethyl ether (7 ml) was added dropwise and the reaction mixture was stirred for an hour at −78°C. Diethyl oxalate (2.11 ml, 15.3 mmol, 1.5 eq.) was added dropwise and the reaction mixture was stirred for 18 h while warming to room temperature. The solvent was evaporated under vacuum and the residue was dissolved in 10% hydrochloric acid (56 ml) and ethyl acetate (56 ml). The organic layers were extracted three times with ethyl acetate, washed with brine, dried over MgSO$_4$, filtered and concentrated to give the crude product as a yellow oil. Product **4** was used in the next step without further purification. $^1$H NMR (400 MHz, CDCl$_3$) $\delta$ ppm 0.97–1.02 (m, 2H, CH$_2$), 1.03–1.08 (m, 2H, CH$_2$), 1.23 (d, 3H, $J$ = 6.2 Hz, CH$_3$CHCO), 1.36 (t, 3H, $J$ = 7.6 Hz, OCH$_2$CH$_3$), 2.01–2.10 (m, 1H, CH), 4.30–4.36 (m, 2H, OCH$_2$CH$_3$), 4.41 (q, 1H, $J$ = 7.2 Hz, CHCH$_3$); $^{13}$C NMR (101 MHz, CDCl$_3$) $\delta$ ppm 14.0 (2x CH$_2$), 14.2 (CH$_3$CH), 17.0 (OCH$_2$CH$_3$), 20.3 (CH), 56.2 (CHCH$_3$), 63.3 (OCH$_2$CH$_3$), 158.0 (C(O)OCH$_2$CH$_3$), 191.2 (CHC(O)C(O)), 207.7 (C(O)CH(CH$_2$)CH$_2$); IR $\nu_{max}$ (thin film, cm$^{-1}$) = 3020, 1733, 1388, 1216, 1193, 1039, 1015, 753, 667; ESI MS $m/z$ [M-H]$^-$: 197.0; HRMS calculated for C$_{10}$H$_{13}$O$_4$ [M-H]$^-$ 197.0806, found 197.0812.

## 4.5. Ethyl 5-cyclopropyl-4-methyl-1H-pyrazole-3-carboxylate

4-Cyclopropyl-3-methyl-2,4-dioxobutanoate **4** (2.06 g, 10.4 mmol, 1.0 eq.) was dissolved in ethanol (42 ml). Hydrazine hydrate (1.07 ml, 20.9 mmol, 62%, 1.0 eq.) and acetic acid (1.19 ml, 20.9 mmol, 2.0 eq.) were added to the reaction and the mixture was allowed to stir for 18 h at 50°C under air.

Ethanol was then removed under vacuum and the organic layers were extracted with dichloromethane. The combined organic layers were washed with brine, dried over MgSO$_4$, filtered and concentrated. Purification by silica gel column chromatography (Hexane/EtOAc 8.5 : 1.5) yielded the title product as a yellow solid (490 mg, 2.26 mmol, 24% yield for three steps). M.p. 67–69°C. $^1$H NMR (400 MHz, CDCl$_3$) $\delta$ ppm 0.80 (d, 2H, $J$ = 4.2 Hz, C$H_2$), 0.92 (d, 2H, $J$ = 7.5 Hz, C$H_2$), 1.39 (t, 3H, $J$ = 6.8 Hz, C$H_3$CH$_2$O), 1.74–1.82 (m, 1H, C$H$), 2.31 (s, 3H, C$H_3$), 4.37 (q, 2H, $J$ = 7.5, 7.1 Hz, CH$_3$C$H_2$O); $^{13}$C NMR (101 MHz, CDCl$_3$) $\delta$ ppm 6.2 (2 x $CH_2$), 6.2 ($CH$), 8.4 ($CH_3$), 14.2 (O$CH_2CH_3$), 60.6 (O$CH_2$CH$_3$), 118.8 ($C$ = C), 160.8 ($C$ = N), 164.8 (HN$C$ = C), 165.0 ($C$(O)); IR $v_{max}$ (thin film, cm$^{-1}$) = 3132, 3022, 2922, 1715, 1630, 1431, 1261, 1215, 1091, 754; ESI MS $m/z$ [M + Na]$^+$: 217.2; HRMS calculated for C$_{10}$H$_{14}$O$_2$N$_2$Na [M + Na]$^+$ 217.0947, found 217.0946.

## 4.6. Ethyl 5-cyclopropyl-1-(4-ethoxy-2,6-difluorobenzyl)-4-methyl-1*H*-pyrazole-3-carboxylate

**7**

Ethyl 5-cyclopropyl-4-methyl-1*H*-pyrazole-3-carboxylate **5** (0.49 g, 2.52 mmol, 1.0 eq.) was dissolved in THF (6 ml) and the solution was cooled to 0°C. Sodium hydride (0.12 g, 3.02 mmol, 60%, 1.2 eq.) was added carefully to the reaction mixture and the solution stirred for 15 min. 2-(Bromomethyl)-5-ethoxy-1,3-difluorobenzene (0.66 g, 2.65 mmol, 1.05 eq.) in THF (6 ml) was then added dropwise at 0°C and the reaction mixture was allowed to stir for 2 h at room temperature. Water (2.8 ml) was added and solvent was removed under vacuum. The aqueous residue was extracted with ethyl acetate and the combined organic layers were washed with brine, dried over MgSO$_4$, filtered and concentrated. Purification by silica gel column chromatography (Hexane/EtOAc 8.5 : 1.5) yielded the title product as a pale yellow solid (690 mg, 1.78 mmol, 75%). M.p. 87–88°C. $^1$H NMR (400 MHz, CDCl$_3$) $\delta$ ppm 0.67 (apparent d, 2H, $J$ = 5.1 Hz, C$H_2$), 0.99 (apparent d, 2H, $J$ = 8.2 Hz, C$H_2$), 1.34–1.42 (m, 6H, 2x C$H_3$CH$_2$O), 1.46–1.53 (m, 1H, C$H$), 2.24 (s, 3H, C$H_3$), 3.97 (q, 2H, $J$ = 7.1 Hz, CH$_3$C$H_2$O), 4.35 (q, 2H, $J$ = 7.4 Hz, CH$_3$C$H_2$O), 5.45 (s, 2H, C$H_2$N), 6.42 (d, 2H, $J$ = 9.7 Hz, ArC$H$); $^{13}$C NMR (126 MHz, CDCl$_3$) $\delta$ ppm 4.7 ($CH$), 5.6 (2x $CH_2$), 9.5 ($CH_3$), 14.4 (O$CH_2CH_3$), 14.5 (O$CH_2CH_3$), 42.3 ($CH_2$N), 60.4 (O$CH_2$CH$_3$), 64.2 (O$CH_2$CH$_3$), 98.4 (d, $J$ = 27.5 Hz, 2x Ar$CH$), 104.1 (d, $J$ = 19.2 Hz, Ar$C$), 119.2 ($C$C = N), 140.1 ($C$C = N), 141.5 ($C$ = C$NCH_2$), 160.4 (t, $J$ = 14.2 Hz, Ar$C$-OCH$_2$CH$_3$), 162.1 (dd, $J$ = 248.7, 11.0 Hz, 2x Ar$C$F), 163.3 ($C$(O)); IR $v_{max}$ (thin film, cm$^{-1}$) = 2987, 1711, 1638, 1585, 1504, 1445, 1345, 1261, 1218, 1147, 1097, 1050, 753, 667; ESI MS $m/z$ [M + Na]$^+$: 387.2; HRMS calculated for C$_{19}$H$_{22}$N$_2$O$_3$F$_2$Na [M + Na]$^+$ 387.1491, found 387.1486.

## 4.7. 5-Cyclopropyl-1-(4-ethoxy-2,6-difluorobenzyl)-4-methyl-1*H*-pyrazole-3-carboxamide hydrochloride

**8**

Ammonium chloride (0.39 g, 7.35 mmol, 5.0 eq.) was suspended in toluene (7 ml) in a sealed tube and trimethyl aluminium solution (2M heptanes, 3.70 ml, 7.35 mmol, 5.0 eq.) was added dropwise at 0°C. The reaction mixture was stirred at room temperature until evolution of gas had stopped. Ethyl 5-cyclopropyl-1-(4-ethoxy-2,6-difluorobenzyl)-4-methyl-1*H*-pyrazole-3-carboxylate **7** (0.53 g, 1.47 mmol, 1.0 eq.) in toluene (5 ml) was added dropwise. The reaction mixture was then stirred for 24 h at 80°C. The reaction was then cooled to 0°C and methanol (35 ml) was added dropwise and the mixture

stirred for an hour at room temperature. The reaction was then filtered through celite and the filter cake was washed with methanol. The solid residue obtained was dried under vacuum and suspended in 50 ml of dichloromethane : methanol (9 : 1). The suspension was stirred for 15 min, filtered and evaporated. The resulting residue was partitioned between dichloromethane and sodium hydrogen carbonate. The organic layers were extracted with dichloromethane, combined, dried over MgSO$_4$ and concentrated to give the title product as a white solid (0.43 g, 1.28 mmol, 78%). M.p. 114–116°C. $^1$H NMR (400 MHz, DMSO-$d_6$) $\delta$ ppm 0.67 (q, 2H, $J$ = 5.8 Hz, C$H_2$), 0.98–1.04 (m, 2H, C$H_2$), 1.31 (t, 3H, $J$ = 7.0 Hz, OCH$_2$C$H_3$), 1.60–1.68 (m, 1H, C$H$), 2.16 (s, 3H, C$H_3$), 4.05 (q, 2H, $J$ = 6.9 Hz, OC$H_2$CH$_3$), 5.31 (s, 2H, C$H_2$N), 6.34–6.59 (s, broad, 2H, N$H_2$), 6.73 (d, 2H, $J$ = 9.9 Hz, 2x ArC$H$); $^{13}$C NMR (126 MHz, MeOH-$d_4$) $\delta$ ppm 4.1 (CH), 4.8 (2x CH$_2$), 7.9 (CH$_3$), 13.4 (OCH$_2$CH$_3$), 40.9 (CH$_2$N), 64.2 (OCH$_2$CH$_3$), 98.0 (d, $J$ = 26.9 Hz, 2 x ArCH,), 103.6 (d, $J$ = 19.7 Hz, ArC), 115.3 (CH$_3$C = C), 139.9 (N = C), 142.9 (C = C-N), 160.7 (NH$_2$C = NH), 161.0 (t, $J$ = 14.4 Hz, ArC-OCH$_2$CH$_3$), 162.2 (dd, $J$ = 246.9, 11.1 Hz, 2x ArCF); IR $\nu_{max}$ (thin film, cm$^{-1}$) = 3478, 3306, 3046, 2982, 1637, 1586, 1505, 1446, 1344, 1266, 1147, 1051, 841, 736, 702; ESI MS $m/z$ [M + H]$^+$: 335.2; HRMS calculated for C$_{17}$H$_{21}$N$_4$OF$_2$ [M + H]$^+$: 335.1678, found 335.1686.

## 4.8. 3,3-Bis(dimethylamino)-2-methoxypropanenitrile

**9**

*Tert*-butoxy *bis*(dimethylamino)methane (4.00 ml, 19.4 mmol, 1.0 eq.) and 2-methoxy acetonitrile (1.44 ml, 19.4 mmol, 1.0 eq.) were stirred in a sealed tube at 80°C for 18 h. The reaction was then concentrated to remove the volatile materials. The crude product was distilled under vacuum to give the product as a yellow liquid (1.98 g, 11.5 mmol, 60%). $^1$H NMR (400 MHz, CDCl$_3$) $\delta$ ppm 2.38 (s, 6H, 2x C$H_3$), 2.40 (s, 6H, 2x C$H_3$), 3.21 (d, 1H, $J$ = 5.5 Hz, ((CH$_3$)$_2$NC$H$N(CH$_3$)$_2$), 3.51 (s, 3H, OC$H_3$), 4.27 (d, 1H, $J$ = 5.5, CHC$H$(OCH$_3$)CN); $^{13}$C NMR (101 MHz, CDCl$_3$) $\delta$ ppm 41.6 (2x CH$_3$), 41.8 (2x CH$_3$), 60.2 (OCH$_3$), 119.5 (C $\equiv$ N), 136.8 (NCH), 143.4 (CHC $\equiv$ N); IR $\nu_{max}$ (thin film, cm$^{-1}$) = 3012, 2182, 1648, 1388, 1216, 1113, 752; ESI $m/z$ [M + H]$^+$: 172.2; HRMS calculated for C$_8$H$_{18}$N$_3$O [M + H]$^+$ 172.1444, found 172.1443.

## 4.9. 2-[5-Cyclopropyl-1-(4-ethoxy-2,6-difluorobenzyl)4-methyl-1H-pyrazol-3-yl]-5-methoxypyrimidin-4-amine

**10**

5-Cyclopropyl-1-(4-ethoxy-2,6-difluorobenzyl)-4-methyl-1H-pyrazole-3-carboxamide hydrochloride **8** (0.20 g, 0.54 mmol, 1.0 eq.) was dissolved in 1-propanol (1.35 ml) in a sealed tube. Piperidine (0.06 ml, 0.59 mmol, 1.1 eq.) was then added dropwise. 3-3-*Bis*(dimethylamino) acetonitrile (0.13 ml, 0.74 mmol, 1.37 eq.) was then added dropwise and the reaction mixture was stirred for 2 min. The mixture was then heated at 100°C for 15 min in the microwave. Purification by silica gel column chromatography (Hexane/EtOAc 9 : 1 to 3 : 7) yielded the title product as a yellow solid (64 mg, 0.15 mmol, 29%). M.p. 150–152°C. $^1$H NMR (400 MHz, DMSO-$d_6$) $\delta$ ppm 0.69 (d, 2H, $J$ = 4.7 Hz, C$H_2$), 1.01 (d, 2H, $J$ = 7.9 Hz, C$H_2$), 1.30 (t, 3H, $J$ = 6.8 Hz, OCH$_2$C$H_3$), 1.63–1.71 (m, 1H, C$H$), 2.22 (s, 3H, C$H_3$), 3.81 (s, 3H, OC$H_3$), 4.04 (q, 2H, $J$ = 7.2 Hz, OC$H_2$CH$_3$), 5.32 (s, 2H, C$H_2$N), 6.74 (d, 2H, $J$ = 9.6 Hz, 2x ArC$H$), 7.84 (s, 1H, ArC$H$ (pyr)); $^{13}$C NMR (126 MHz, DMSO-$d_6$) $\delta$ ppm 5.1 (CH), 5.7 (2x CH$_2$), 10.5 (CH$_3$), 14.8 (OCH$_2$CH$_3$), 40.6 (NCH$_2$), 56.1 (OCH$_3$), 64.7 (OCH$_2$CH$_3$), 99.0 (d, $J$ = 24.9 Hz, 2 x ArCH), 105.3 (t, $J$ = 19.4 Hz, ArC), 114.5 (CH$_3$C = C), 133.7 (ArCH (pyr)), 138.6 (ArC-OCH$_3$), 140.9 (C = C-N), 147.6 (C = N in pyrazole), 154.1 (ArC-NH$_2$), 155.3 (N = C-N in pyr), 160.3 (t, $J$ = 14.3 Hz, ArC-OCH$_2$CH$_3$), 162.1 (dd,

$J$ = 246.1, 11.9 Hz, 2x ArCF); IR $\nu_{max}$ (thin film, cm$^{-1}$) = 3490, 3282, 3163, 3046, 3047, 2980, 2930, 1636, 1583, 1503, 1490, 1445, 1341, 1265, 1234, 1147, 1049, 738; ESI MS $m/z$ [M + Na]$^+$: 438.3; HRMS calculated for $C_{21}H_{24}N_5O_2F_2$ [M + H]$^+$: 416.1893, found 416.1887.

## 4.10. 2-[5-Cyclopropyl-1-(4-ethoxy-2,6-difluorobenzyl)-4-methyl-1*H*-pyrazol-3-yl]-5-methoxy-*N*-(pyridine-4-yl)pyrimidin-4-amine

**11**

2-[5-Cyclopropyl-1-(4-ethoxy-2,6-difluorobenzyl)4-methyl-1*H*-pyrazol-3-yl]-5-methoxypyrimidin-4-amine **10** (13 mg, 0.03 mmol, 1.0 eq.) was added to a sealed tube and 4-iodopyridine (10 mg, 0.05 mmol, 1.5 eq.), Cs$_2$CO$_3$ (41 mg, 0.12 mmol, 4.0 eq.), Xantphos (3.6 mg, 0.006 mmol, 0.2 eq.) and palladium(II) acetate (2.1 mg, 0.01 mmol, 0.3 eq.) were added and were suspended in DMF (0.3 ml). The resulting mixture was stirred for 24 h at 110°C. The mixture was then partitioned between dichloromethane and saturated ammonium chloride. The organic layers were extracted with dichloromethane, combined, washed with brine, dried over MgSO$_4$, filtered and concentrated. Purification by silica gel column chromatography (Hexane/EtOAc 9 : 1 to 2 : 8) yielded the title product as a white solid (10 mg, 0.019 mmol, 68%). M.p. 50–52°C. $^1$H NMR (400 MHz, DMSO-$d_6$) $\delta$ ppm 0.75 (d, 2H, $J$ = 5.5 Hz, C*H*$_2$), 1.07 (d, 2H, $J$ = 8.9 Hz, C*H*$_2$), 1.31 (t, 3H, $J$ = 6.9 Hz, OCH$_2$C*H*$_3$), 1.73 (apparent td, 1H, $J$ = 8.5, 4.3 Hz, C*H*), 2.30 (s, 3H, C*H*$_3$), 3.98 (s, 3H, OC*H*$_3$), 4.00–4.06 (m, 2H, OC*H*$_2$CH$_3$), 5.37 (s, 2H, C*H*$_2$N), 6.79 (d, 2H, $J$ = 11.2 Hz, ArC*H*), 8.10 (d, 2H, $J$ = 6.1 Hz, 2x ArC*H* (py)), 8.22 (s, 1H, ArC*H* (pyr)), 8.33 (d, 2H, $J$ = 7.4 Hz, 2x ArC*H* (py)); $^{13}$C NMR (101 MHz, MeOD-$d_4$) $\delta$ ppm 4.4 (C*H*), 4.8 (2x *C*H$_2$), 9.0 (*C*H$_3$), 13.4 (OCH$_2$*C*H$_3$), 40.5 (*C*H$_2$), 55.5 (O*C*H$_3$), 64.1 (O*C*H$_2$CH$_3$), 98.0 (d, $J$ = 29.3 Hz, 2x Ar*C*H,), 104.4 (d, $J$ = 19.3 Hz, Ar*C*), 114.2 (2x Ar*C*H (py)), 114.8 (CH$_3$*C*=C), 134.4 (Ar*C*H (pyr)), 139.7 (*C*=N (pyrazole)), 141.8 (*C*=*C*-N), 147.1 (Ar*C*=N (pyr)), 147.6 (Ar*C* (py)), 148.7 (2x Ar*C*H (py)), 150.6 (Ar*C* (pyr)), 152.9 (Ar*C*-NH (pyr)), 160.9 (d, $J$ = 33.6 Hz, Ar*C*-OCH$_2$CH$_3$), 2x Ar*C*F were not observed; IR $\nu_{max}$ (thin film, cm$^{-1}$) = 2925, 2851, 1639, 1604, 1577, 1504, 1461, 1342, 1265, 1146, 1052, 738; ESI MS $m/z$ [M + H]$^+$: 493.3; HRMS calculated for $C_{26}H_{26}N_6O_2F_2Na$ [M + Na]$^+$ 515.1978, found 515.1967.

## 4.11. Materials and cell lines

Tetracycline hydrochloride was dissolved in water and used at 1 µg ml$^{-1}$. Small-molecule inhibitors, 2OH-BNPP1 (Peakdale) and BAY-320, were dissolved in DMSO and used at 10 µM, unless stated otherwise. Nocodazole was also dissolved in DMSO and used at 200 ng ml$^{-1}$ (Sigma Aldrich). Nutlin-3 was used at 10 µM in DMSO.

The human colon carcinoma cell lines RKO1, RKO1 *TP53*$^{-/-}$, DLD-1 expressing GFP-H2B [44], HEK-293, HeLa cell lines and their Flp-In$^{TM}$ T-REx$^{TM}$ derivatives (see below) and the RPE1 cells were all cultured in Dulbecco's modified Eagle's medium (DMEM, Invitrogen) supplemented with 10% fetal bovine serum (Gibco), 100 U ml$^{-1}$ penicillin, 100 U ml$^{-1}$ streptomycin and 2 mM glutamine (all Sigma Aldrich) and maintained at 37°C in a humidified 5% CO$_2$ atmosphere; note that pre-transfection, media was supplemented with blasticidin (DLD-1, 8 µg ml$^{-1}$; HEK-293, 15 µg ml$^{-1}$; HeLa, 4 µg ml$^{-1}$; Melford Laboratories) and zeocin (DLD-1, 60 µg ml$^{-1}$; HEK-293, 100 µg ml$^{-1}$; HeLa, 50 µg ml$^{-1}$; Sigma Aldrich). Established ovarian carcinoma cell lines OVCAR-3 (ATCC) and Kuramochi (JCRB Cell Bank) were cultured in RPMI (Invitrogen) supplemented with 10% fetal bovine serum (Gibco), 100 U ml$^{-1}$ penicillin, 100 U ml$^{-1}$ streptomycin and 2 mM glutamine (all Sigma Aldrich) and maintained at 37°C in a humidified 5% CO$_2$ atmosphere. All lines were authenticated by the Molecular Biology Core Facility at the CRUK Manchester Institute using Promega Powerplex21 System and periodically tested for mycoplasma.

HEK-293 and HeLa cell lines expressing tetracycline-inducible exogenous Bub1 fusions, GFP-Bub1 and GFP-H2B-Bub1C, respectively, were created using the Flp-In™ T-REx system (Invitrogen). Full-length *BUB1* and the *BUB1C* allele (region encoding amino acids 724–1085) were PCR amplified with *Bam*HI and *Not*I sites in the forward and reverse primers and the products cloned into pcDNA5/FRT/TO (Invitrogen) containing GFP or GFP-H2B, respectively. Plasmids were transformed into XL1-Blue competent cells and plasmid DNA extracted using QIAprep Spin Miniprep Kit (Qiagen). The *BUB1 K821R* and *BUB1C D946N* alleles were generated by single-base substitution using site-directed mutagenesis (Strategene) on pcDNA5/FRT/TO-GFP-Bub1 and pcDNA5/FRT/TO-GFP-H2B-Bub1C, respectively. Mutagenesis primers were: K821R-F, 5′ GAT GCT AAA AAT AAA CAG AAA TTT GTT TTA AGG GTC CAA AAG CCT GCC 3′; K821R-R, 5′ GGC AGG CTT TTG GAC CCT TAA AAC AAA TTT CTG TTT ATT TTT AGC ATC 3′; D946N-F, 5′ TCT GCT GGC TTG GCA CTG ATT AAC CTG GGT CAG 3′; D946N-R, 5′ CTG ACC CAG GTT AAT CAG TGC AAG CCA GCA GAA 3′. Subsequent cloned vectors were co-transfected with pOG44 into Flp-In™ T-REx™ cells. Following selection in either 150 µg ml$^{-1}$ hygromycin B and 15 µg ml$^{-1}$ blasticidin for HEK-293 cells or 200 µg ml$^{-1}$ hygromycin B and 4 µg ml$^{-1}$ blasticidin for HeLa cells, colonies were pooled and expanded to create an isogenic polyclonal cell line. All constructs were confirmed by full sequencing. Expression of Bub1 clones by the addition of tetracycline hydrochloride was confirmed by immunoblotting.

For CRISPR-Cas9-mediated mutagenesis to generate *TP53*$^{-/-}$ RKO1 cells, $1.6 \times 10^5$ RKO cells per well were seeded under in a 24-well plate (Corning) and maintained at 37°C in a humidified 5% $CO_2$ atmosphere overnight. Transfection of a pD1301-based plasmid (Horizon Discovery), which expresses Cas9, an EmGFP-tag and a sgRNA targeting *TP53* (5′ AAT GTT TCC TGA CTC AGA GG 3′), was performed using Lipofectamine 2000, according to manufacturer's instructions. After incubating in DMEM at 37°C in a humidified 5% $CO_2$ for 48 h, transfected cells were sorted by flow cytometry using a BD Influx™ cell sorter and GFP-positive cells seeded one cell per well in 96-well plates (Corning) to generate monoclonal cell lines. Clonal lines were screened by immunoblotting to identify desired clones.

## 4.12. Co-immunoprecipitation

The GST-GFP-binder protein was used for affinity purification of GFP-Bub1 proteins [53,54]. The open reading frame of the GFP-binder protein was cloned into pGEX-4T3 vector and transformed into BL21 cells. GST-GFP-binder expression was induced with IPTG and glutathione sepharose beads (Amintra) used to purify the fusion protein with glutathione used for elution. Cells were incubated with nocodazole and tetracycline for 16–18 h to induce expression of exogenous GFP-Bub1 proteins. After growing to near 100% confluency, cells were harvested and pellets resuspended in lysis buffer (0.1% Triton X-100, 100 mM NaCl, 10 mM Tris pH 7.4, 1 mM EGTA, 20 mM beta-glycerol, 10 mM NaF), cOmplete Mini, EDTA-free Proteasome Inhibitor Cocktail Tablet (Roche) and phosphatase inhibitor tablet (PhosSTOP EASYpack, Roche) and incubated at 4°C for 20 min before centrifugation to remove insoluble proteins. Glutathione sepharose beads (Amintra) were washed twice in lysis buffer and incubated with the soluble fraction and purified GST-GFP-binder protein at 4°C for at least 3 h. Co-immunoprecipitation beads were washed five times in lysis buffer and used directly for immunoblotting or kinase assays.

## 4.13. Immunoblotting

Proteins were extracted by boiling samples or beads in sample buffer (0.35 M Tris pH 6.8, 0.1 g ml$^{-1}$ sodium dodecyl sulfate, 93 mg ml$^{-1}$ dithiothreitol, 30% glycerol, 50 µg ml$^{-1}$ bromophenol blue), resolved by SDS-PAGE, then electroblotted onto Immobilon-P membranes (Merck Millipore). Following blocking in 5% dried skimmed milk (Marvel) dissolved in TBST (50 mM Tris pH 7.6, 150 mM NaCl, 0.1% Tween-20), membranes were incubated overnight at 4°C with primary antibodies: sheep anti-Bub1 (SB1.3) [42] (1 : 1000); rabbit anti-H2ApT120 (Active Motif cat#39391, 1 : 1000); sheep anti-Bub3 (SB3.2; 1 : 1000)); rabbit anti-GFP (Cell Signaling cat#2956, 1 : 1000); sheep anti-Tao1 [55] (1 : 1000); mouse anti-p53 (DO-1) (Santa Cruz Biotechnology cat#sc-126, 1 : 1000). Membranes were then washed three times in TBST and incubated for at least 1 h with appropriate horseradish-peroxidase-conjugated secondary antibodies (1 : 2000). After washing in TBST, bound secondary antibodies were detected using either EZ-Chemiluminescence Reagent (Geneflow Ltd) or Luminata™ Forte Western HRP Substrate (Merck Millipore) and a Biospectrum 500 imaging system (UVP) or a ChemiDoc™ Touch Imaging System (BioRad). To process the images VisionWorks®LS (UVP) was used.

## 4.14. *In vitro* kinase assays

In the early stages of the study, Bub1 kinase activity was measured using radiolabelled ATP. Washed glutathione sepharose beads with bound GFP-Bub1 following co-immunoprecipitation were subsequently washed three times in kinase buffer (25 mM Tris-HCl pH 7.4, 100 mM NaCl, 10 mM MgCl$_2$, 50 µg ml$^{-1}$ bovine serum albumin [Sigma], 0/1 mM EGTA, 0.1% $\beta$-mercaptoethanol). Beads (10 µl) were mixed with 100 µM ATP and 2 µg H2A (New England Biosciences) on ice before incubation at 30°C for 20 min. Following reaction completion, proteins were denatured by boiling in sample buffer and resolved by SDS-PAGE. $\lambda$-$^{32}$P-ATP (2 µCi; PerkinElmer) was included in reactions to enable measurement of phosphorylation during assay optimization using a phosphorimager (Typhoon FLA7000, Raytek Scientific Limited, Sheffield, UK). ImageJ was used for quantification. In the latter phase of the project, radiolabelling assays were no longer possible due to regulatory constraints following relocation of the laboratory to a new building, so Bub1 kinase activity was measured by immunoblotting for H2ApT120 (see above).

## 4.15. Immunofluorescence microscopy

Cell lines were plated onto 19 mm coverslips (VWR International) approximately 24 h prior to drug treatment. Cells were treated with nocodazole ± tetracycline for 16–18 h, followed by 3 h 2OH-BNPP1 or BAY-320 as indicated in figure legends. Cells were then washed and fixed in 1% formaldehyde, quenched in glycine (1M glycine pH 8.5 with 1M Tris pH 8.5) and permeabilized with PBS-T (PBS with 0.1% Triton X-100) before incubation for 30 min at room temperature with primary antibodies: sheep anti-Bub1 (SB1.3) [42] (1 : 1000); rabbit anti-H2ApT120 (Active Motif cat#3939, 1 : 1000); sheep anti-Sgo1 (1 : 1000); human anti-ACA (1 : 1000). Coverslips were washed two times in PBS-T and incubated with the appropriate fluorescently conjugated secondary antibodies (1 : 500) for 30 min at room temperature. Coverslips were washed in PBS-T and DNA stained for 1 min with 1 µg ml$^{-1}$ Hoechst 33258 (Sigma) at room temperature. Coverslips were further washed in PBS-T and mounted (90% glycerol, 20 mM Tris, pH 8.0) onto slides. Image acquisition was done using an Axioskop2 (Zeiss, Inc.) microscope with a 32x or 100x objective fitted with a CoolSNAP HQ camera (Photometrics) with analysis of images performed using MetaMorph Software (Molecular Devices). Deconvolution microscopy (DeltaVision RT; Applied Precision) was performed using a 100x 1.40 NA Plan Apo objective and filter set (Sedat Quad; Chroma Technology Corp). Image analysis was conducted using Adobe Photoshop® CC 2015 (Adobe Systems Inc.). For high-throughput immunofluorescence, cells were processed as above in 96-well plate format (PerkinElmer Cell Carrier plates) and stored in PBS at 4°C prior to imaging. Images were acquired using Operetta® High Content Imaging System (PerkinElmer) and quantified using Harmony and Columbus High Content Imaging and Analysis Software (PerkinElmer) to measure fluorescence intensity.

## 4.16. Time-lapse microscopy

Cells were plated at $8 \times 10^4$ cells ml$^{-1}$ in a 96-well plate (Corning). After drug addition, time-lapse microscopy was performed on an Axiovert 200 manual microscope (Zeiss, Inc.) equipped with an automated stage (PZ-2000; Applied Scientific Instrumentation) and an environmental control chamber (Solent Scientific), which maintained the cells at 37°C in a humidified stream of 5% CO$_2$. Imaging was performed using a 40x Plan NEOFLUAR objective. Shutters, filter wheels and point visiting were driven by MetaMorph software (Molecular Devices). Images were taken using an Evolve delta camera (Photometrics). Images were processed using Photoshop (Adobe) and Quicktime (Apple). To measure time in mitosis, nuclear envelope breakdown was judged as the point when the prophase chromatin lost a smooth, linear periphery, and the time of anaphase onset was judged to be first frame where coordinated polewards movement was observed.

## 4.17. Cell fate profiling

Cells were seeded at $8 \times 10^4$ cells ml$^{-1}$ in 96-well plates (Greiner Bio-One), 24 h prior to drug treatment. Cells were imaged using an IncuCyte® ZOOM (Essen BioScience) equipped with a 20x objective and maintained at 37°C in a humidified 5% CO$_2$ atmosphere. Nine phase contrast and fluorescence images per well were collected every 10 min for 72 h for cell fate profiling. For cell fate profiling, image

sequences were exported in MPEG-4 format and analysed manually to generate cell fate profiles. Timing data were imported into Prism 7 (GraphPad) for presentation.

## 4.18. Colony formation assay

For colony formation assays 2000 cells/well were seeded into 6-well plates and treated with the inhibitors for 72 h then washed out. Once colonies had developed, the cells were fixed in 1% formaldehyde and stained with 0.05% (w/v) crystal violet solution. Plates were then imaged using a ChemiDocTM Touch Imaging System (BioRad).

## 4.19. Mouse embryonic fibroblast experiments

The *Bub1 K795R* allele was generated by a single-base substitution in full-length mouse *Bub1* cDNA cloned into pcDNA3/Myc using site-directed mutagenesis (Stratagene), then subcloned into a pShuttleCMV vector as a *Bgl*II-*Not*I digest. pShuttleCMV-Bub1K795R was used to generate recombinant adenoviruses using the AdEasy system (Stratagene), according to the manufacturer's instructions. Adenoviruses containing wild-type Bub1, and the ΔKD, D919N and Δ38 variants, along with immortalized MEFs harbouring tamoxifen-responsive Cre recombinase and a single *BUB1* allele floxed between two lox P sites (*BUB1*$^{F/Δ}$) were created previously in our laboratory [14]. Immortalized MEF cultures were infected with the adenoviruses with a multiplicity of infection of approximately 100. To activate Cre, MEFs were cultured in optiMEM media (Invitrogen) plus 10% charcoal-dextran-treated serum (Hyclone) with OHT. For time-lapse microscopy, monastrol-treated cells were seeded in 30 mm glass-bottomed Petri dishes (MatTek Co) and then transferred to the microscope stage. Images were taken every 2 min for up to 24 h, with time in mitosis defined as described above. Monastrol (Sigma Aldrich) was used at a final concentration of 100 μM and OHT dissolved in ethanol was used at a final concentration of 0.5 μM.

## 4.20. Statistical analysis

Time in mitosis was analysed using Microsoft Excel and graphs were created with paired *t*-test using GraphPad Prism 7. Statistical analyses were performed with the non-parametric Mann–Whitney U Tests using GraphPad Prism 6. The box plots show the mean and interquartile range. Error bars show the standard deviation. Note that $^{****}p < 0.0001$, $^{***}p < 0.001$, $^{**}p < 0.01$, $^{*}p < 0.05$, ns: $p > 0.05$.

Data accessibility. Datasets supporting this article are available in the electronic supplementary material [56].

Authors' contributions. A.B. established the *in vitro* and cell-based Bub1 kinase assays. I.A. synthesized BAY-320 and performed the comparative analysis and cell biology experiments. D.P. contributed the MEF data, and H.W. contributed the RKO *TP53*$^{-/-}$ cell line. A.T. provided lab supervision and microscopy support. S.S.T. analysed all the data and created the figures. J.C.M. wrote the manuscript. The project was conceived and overseen by D.J.P and S.S.T. All authors reviewed the manuscript and gave final approval for publication and agree to be held accountable for the work performed therein.

Competing interests. The authors declare no competing interests.

Funding. I.A. was supported by the Indonesia Endowment Fund for Education (LPDP Scholarship). A.B. and research in the Taylor lab were funded by Cancer Research UK Programme Grant to S.S.T (grant no. C1422/A19842).

Acknowledgements. We thank the members of the Taylor and Procter labs for advice and comments on the manuscript.

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
