## [Peer Review File · Royal Society Open Science]

Review History

RSOS-210854.R0 (Original submission)

Review form: Reviewer 1

Is the manuscript scientifically sound in its present form?

Yes

Are the interpretations and conclusions justified by the results?

Yes

Is the language acceptable?

Yes

Do you have any ethical concerns with this paper?

No

Have you any concerns about statistical analyses in this paper?

No

Recommendation?

Accept as is

Comments to the Author(s)

Amalina et al. have produced a detailed study of two compounds purported to be selective inhibitors on Bub1 kinase. This study is important because - as is well explained in the introduction - the role of Bub1, specifically its kinase activity is contentious. Selective inhibitor(s) would help the field to get to the bottom of some of these discrepancies. What is presented here is a very carefully done study which confirms the in vitro inhibition of two compounds and tests the action of these compounds in cellulo, in part using a neat genetically encoded reporter of Bub1 activity. The results essentially underline that more development is required to obtain selective Bub1 kinase inhibitors.

The work is very high quality, and in my opinion scientifically sound. Note that I am not qualified to judge the chemistry aspects (synthesis of BAY-320). It is a useful contribution to a controversial area of cancer cell biology. I have some very minor comments that the authors can address straightforwardly.

Figure 7 the 10 μ M BAY-320 experiments need a time scale on the x-axis.

Figure 7 the legend says 100 cells were analysed per condition. How are these selected? Random or some other method? In addition, if this is a single overnight imaging experiment (or a composite of different replicates) it should be stated in the legend.

Figure 2 is the compression of 1.5-2 on the y-axis of the inhibition graph for 2OH-BNPP1 intentional?

line 299 "to H2B to ectopically delocalised Bub1 kinase" -> "to H2B to ectopically delocalise Bub1 kinase"

Line 717 Ms. Excel -> Microsoft Excel

Review form: Reviewer 2

Is the manuscript scientifically sound in its present form?

Yes

Are the interpretations and conclusions justified by the results?

Yes

Is the language acceptable?

Yes

Do you have any ethical concerns with this paper?

No

Have you any concerns about statistical analyses in this paper?

No

Recommendation?

Accept with minor revision (please list in comments)

Comments to the Author(s)

There has been significant recent controversy in the literature regarding the role of Bub1 in mitosis. Conflicting results have been obtained using siRNA-mediated knockouts, CRISPR-mediated deletions, and small molecule-mediated inhibition. As there is significant clinical interest in Bub1 as a cancer target, it is important to develop an accurate picture of its cellular function. This new paper from the Taylor lab makes an excellent contribution toward that overall goal. Here, the authors carefully compare the effects of two putative Bub1 inhibitors in a series of *in vitro* and *in vivo* assays. They conclude that one inhibitor, BAY-320, is more potent against Bub1 *in vivo*, but they caution that it may exhibit some degree of promiscuous off-target binding. This is exactly the type of careful work that should be published.

I have two brief suggestions for improvement:

1) In certain locations, the authors make blanket statements about the effects of the 2OH drug, e.g., "BAY-320, but not 2OH-BNPP1, inhibits Bub1 *in vivo*". The authors tested the 2OH drug at concentrations up to 10uM, but it is formally possible that they could see an inhibitory effect at 11uM, 20uM, or 200uM. Because of this consideration, I think that it is not appropriate to explicitly say "2OH-BNPP1 does not inhibit Bub1 *in vivo*". The authors should temper this statement with modifiers - "does not effectively inhibit", "does not inhibit at the concentrations tested", etc., as they already do in several locations.

2) The key experiment to demonstrate off-target effects of a mitotic kinase inhibitor is to treat cells carrying a genetic knockout of that drug's target with the drug itself (as done in for instance PMID: 29417930). If the authors hypothesize that 10uM treatment with BAY-320 has promiscuous off-target effects, then it would seem feasible to show that this concentration is also cytotoxic in the BUB1-deletion MEFs that the authors have generated. Please note: while I think that this experiment would strengthen the manuscript, if it is not feasible due to the time required for 4-OHT-mediated Bub1 disruption or for some other reason, I would still support publication without that experiment.

Review form: Reviewer 3

Is the manuscript scientifically sound in its present form?

Yes

Are the interpretations and conclusions justified by the results?

Yes

Is the language acceptable?

Yes

Do you have any ethical concerns with this paper?

No

Have you any concerns about statistical analyses in this paper?

No

Recommendation?

Accept with minor revision (please list in comments)

Comments to the Author(s)

Amalina et al evaluate two small molecule inhibitors (2OH-BNPP1 and BAY-320) of the kinase Bub1 using in vitro and cell-based assays.

They find that in vitro both compounds inhibit Bub1 efficiently. However, in cells, only BAY-320 inhibits exogenous and endogenous Bub1 kinase activity. Because 4% remaining of Bub1 is sufficient for spindle checkpoint function, a concentration of BAY-320 is needed to fully inhibit Bub1 for functional analysis. At this concentration, the authors observe chromosome alignment defects, delayed mitotic progression, and cell death. Because cell death is mainly observed in interphase, the authors are concerned about potential off-target effects.

This study describes a series of well-designed and carefully carried out experiments clarifying the role of two Bub1 inhibitors. Small molecule kinase inhibitors are critical therapeutics in anti-cancer treatment and tools for biological investigations. Thus, careful characterization of their on- and off-target effects is essential, and this study makes an essential contribution to the evaluation of these compounds.

Minor points:

1. Why did the authors switch from 32P to Western blotting for the inhibitor testing? Please include an explanation.
2. The authors use two different kinase-dead mutations (D946N and K821R/K795R), but only test K821R in vitro and D946N in cells. These results should be reconciled.
3. Why did the authors pretreat DLD-1 cells for 3 days with BAY-320 before evaluating mitotic duration? In the analysis of endogenous H2AT210 phosphorylation and Sgo1 localization, the authors only treated for 3 hrs? How was mitotic timing affected by shorter treatments? What was the mitotic index in cells treated for 3 days with BAY-320 compared to untreated or control-treated cells?
4. Why do the authors use various cell lines throughout the study? Please explain. It would be more informative if all experiments (analysis of chromosome alignment, SAC activity, mitotic timing, colony formation, and long-term survival) would be carried out consistently in at least one cell line, and others would be added as needed.

Decision letter (RSOS-210854.R0)

Dear Dr Taylor

On behalf of the Editors, we are pleased to inform you that your Manuscript RSOS-210854 "Inhibitors of the Bub1 spindle assembly checkpoint kinase: Synthesis of BAY-320 and comparison with 2OH-BNPP1" has been accepted for publication in Royal Society Open Science subject to minor revision in accordance with the referees' reports. Please find the referees' comments along with any feedback from the Editors below my signature.

Please submit your revised manuscript and required files (see below) no later than 7 days from today's (ie 28-Oct-2021) date. Note: the ScholarOne system will 'lock' if submission of the revision

is attempted 7 or more days after the deadline. If you do not think you will be able to meet this deadline please contact the editorial office immediately.

on behalf of Dr Simon Cook (Associate Editor) and Catrin Pritchard (Subject Editor)
openscience@royalsociety.org

Associate Editor Comments to Author (Dr Simon Cook):

Comments to the Author:

This is a careful, thorough and excellent study that I fully expect to see published. However, each referee has raised some minor queries that I think should be addressed prior to acceptance

Reviewer comments to Author:

Reviewer: 1

Comments to the Author(s)

Amalina et al. have produced a detailed study of two compounds purported to be selective inhibitors on Bub1 kinase. This study is important because - as is well explained in the introduction - the role of Bub1, specifically its kinase activity is contentious. Selective inhibitor(s) would help the field to get to the bottom of some of these discrepancies. What is presented here is a very carefully done study which confirms the in vitro inhibition of two compounds and tests the action of these compounds in cellulo, in part using a neat genetically encoded reporter of Bub1 activity. The results essentially underline that more development is required to obtain selective Bub1 kinase inhibitors.

The work is very high quality, and in my opinion scientifically sound. Note that I am not qualified to judge the chemistry aspects (synthesis of BAY-320). It is a useful contribution to a controversial area of cancer cell biology. I have some very minor comments that the authors can address straightforwardly.

Figure 7 the 10 μ M BAY-320 experiments need a time scale on the x-axis.

Figure 7 the legend says 100 cells were analysed per condition. How are these selected? Random or some other method? In addition, if this is a single overnight imaging experiment (or a composite of different replicates) it should be stated in the legend.

Figure 2 is the compression of 1.5-2 on the y-axis of the inhibition graph for 2OH-BNPP1 intentional?

line 299 "to H2B to ectopically delocalised Bub1 kinase" -> "to H2B to ectopically delocalise Bub1 kinase"

Line 717 Ms. Excel -> Microsoft Excel

Reviewer: 2

Comments to the Author(s)

There has been significant recent controversy in the literature regarding the role of Bub1 in mitosis. Conflicting results have been obtained using siRNA-mediated knockouts, CRISPR-mediated deletions, and small molecule-mediated inhibition. As there is significant clinical interest in Bub1 as a cancer target, it is important to develop an accurate picture of its cellular function. This new paper from the Taylor lab makes an excellent contribution toward that overall goal. Here, the authors carefully compare the effects of two putative Bub1 inhibitors in a series of *in vitro* and *in vivo* assays. They conclude that one inhibitor, BAY-320, is more potent against Bub1 *in vivo*, but they caution that it may exhibit some degree of promiscuous off-target binding. This is exactly the type of careful work that should be published.

I have two brief suggestions for improvement:

- 1) In certain locations, the authors make blanket statements about the effects of the 2OH drug, e.g., "BAY-320, but not 2OH-BNPP1, inhibits Bub1 *in vivo*". The authors tested the 2OH drug at concentrations up to 10uM, but it is formally possible that they could see an inhibitory effect at 11uM, 20uM, or 200uM. Because of this consideration, I think that it is not appropriate to explicitly say "2OH-BNPP1 does not inhibit Bub1 *in vivo*". The authors should temper this statement with modifiers - "does not effectively inhibit", "does not inhibit at the concentrations tested", etc., as they already do in several locations.
- 2) The key experiment to demonstrate off-target effects of a mitotic kinase inhibitor is to treat cells carrying a genetic knockout of that drug's target with the drug itself (as done in for instance PMID: 29417930). If the authors hypothesize that 10uM treatment with BAY-320 has promiscuous off-target effects, then it would seem feasible to show that this concentration is also cytotoxic in the BUB1-deletion MEFs that the authors have generated. Please note: while I think that this experiment would strengthen the manuscript, if it is not feasible due to the time required for 4-OHT-mediated Bub1 disruption or for some other reason, I would still support publication without that experiment.

Reviewer: 3

Comments to the Author(s)

Amalina et al evaluate two small molecule inhibitors (2OH-BNPP1 and BAY-320) of the kinase Bub1 using *in vitro* and cell-based assays.

They find that *in vitro* both compounds inhibit Bub1 efficiently. However, in cells, only BAY-320 inhibits exogenous and endogenous Bub1 kinase activity. Because 4% remaining of Bub1 is sufficient for spindle checkpoint function, a concentration of BAY-320 is needed to fully inhibit Bub1 for functional analysis. At this concentration, the authors observe chromosome alignment defects, delayed mitotic progression, and cell death. Because cell death is mainly observed in interphase, the authors are concerned about potential off-target effects.

This study describes a series of well-designed and carefully carried out experiments clarifying the role of two Bub1 inhibitors. Small molecule kinase inhibitors are critical therapeutics in anti-cancer treatment and tools for biological investigations. Thus, careful characterization of their on- and off-target effects is essential, and this study makes an essential contribution to the evaluation of these compounds.

Minor points:

1. Why did the authors switch from 32P to Western blotting for the inhibitor testing? Please include an explanation.
2. The authors use two different kinase-dead mutations (D946N and K821R/K795R), but only test K821R *in vitro* and D946N in cells. These results should be reconciled.

3. Why did the authors pretreat DLD-1 cells for 3 days with BAY-320 before evaluating mitotic duration? In the analysis of endogenous H2AT210 phosphorylation and Sgo1 localization, the authors only treated for 3 hrs? How was mitotic timing affected by shorter treatments? What was the mitotic index in cells treated for 3 days with BAY-320 compared to untreated or control-treated cells?

4. Why do the authors use various cell lines throughout the study? Please explain. It would be more informative if all experiments (analysis of chromosome alignment, SAC activity, mitotic timing, colony formation, and long-term survival) would be carried out consistently in at least one cell line, and others would be added as needed.

===PREPARING YOUR MANUSCRIPT===

one version should clearly identify all the changes that have been made (for instance, in coloured highlight, in bold text, or tracked changes);

===PREPARING YOUR REVISION IN SCHOLARONE===

-- If you are requesting an article processing charge waiver, you must select the relevant waiver option (if requesting a discretionary waiver, the form should have been uploaded, see 'File upload' above).

-- If you have uploaded any electronic supplementary (ESM) files, please ensure you follow the guidance at <https://royalsociety.org/journals/authors/author-guidelines/#supplementary-material> to include a suitable title and informative caption. An example of appropriate titling and captioning may be found at https://figshare.com/articles/Table_S2_from_Is_there_a_trade-off_between_peak_performance_and_performance_breadth_across_temperatures_for_aerobic_scope_in_teleost_fishes_/3843624.

Author's Response to Decision Letter for (RSOS-210854.R0)

See Appendix A.

Decision letter (RSOS-210854.R1)

Dear Dr Taylor,

I am pleased to inform you that your manuscript entitled "Inhibitors of the Bub1 spindle assembly checkpoint kinase: Synthesis of BAY-320 and comparison with 2OH-BNPP1" is now accepted for publication in Royal Society Open Science.

on behalf of Dr Simon Cook (Associate Editor) and Catrin Pritchard (Subject Editor)
openscience@royalsociety.org

Dr Simon Cook

Editor, *Royal Society Open Science*
Publishing Section, The Royal Society,
6-9 Carlton House Terrace,
London SW1Y 5AG
15th Nov 2021

Manuscript ID RSOS-210854

Dear Dr Cook,

Please find enclosed the revised version of our manuscript '*Inhibitors of the Bub1 spindle assembly checkpoint kinase: Synthesis of BAY-320 and comparison with 2OH-BNPP1*'. We thank the reviewers for taking the time to evaluate our submission in detail, and we were of course delighted with their overall positive response. On the following pages is a point-by-point response to the individual comments raised, but in summary we have:

- Shortened the abstract to 200 words and added a 100-word media summary, in line with journal guidelines
- Toned down language throughout to state that 2OH-BNPP1 did not *effectively* inhibit Bub1 *at the concentrations tested*
- Made a number of other minor changes to address the reviewer's comments (highlighted)

The net result of these changes is that in our view this is now ready for publication in *Royal Society Open Science*. As The University of Manchester has a Read & Publish agreement with Royal Society Publishing we appreciate that there is no Article Processing Charge payable for publishing this manuscript.

I look forward to hearing from you.

Yours sincerely,

Stephen Taylor, Leech Professor of Pharmacology &
Head of Division for Cancer Sciences
T: 0161 306 0869 or 0161 306 0078 (PA)
E: stephen.taylor@manchester.ac.uk
W: www.bub1.com

Response to reviewers

Dear Dr Taylor

On behalf of the Editors, we are pleased to inform you that your Manuscript RSOS-210854 "Inhibitors of the Bub1 spindle assembly checkpoint kinase: Synthesis of BAY-320 and comparison with 2OH-BNPP1" has been accepted for publication in Royal Society Open Science subject to minor revision in accordance with the referees' reports. Please find the referees' comments along with any feedback from the Editors below my signature.

Please submit your revised manuscript and required files (see below) no later than 7 days from today's (ie 28-Oct-2021) date. Note: the ScholarOne system will 'lock' if submission of the revision is attempted 7 or more days after the deadline. If you do not think you will be able to meet this deadline please contact the editorial office immediately.

on behalf of Dr Simon Cook (Associate Editor) and Catrin Pritchard (Subject Editor)
openscience@royalsociety.org

Associate Editor Comments to Author (Dr Simon Cook):

Comments to the Author:

This is a careful, thorough and excellent study that I fully expect to see published. However, each referee has raised some minor queries that I think should be addressed prior to acceptance

We thank Dr Cook for reviewing the manuscript and supporting publication, we have addressed or responded to each minor point raised by the reviewers below. We have also shortened the abstract to 200 words and added a media summary to align with journal guidelines. In addition, we preferred the reviewer's use of *in cellulo*, rather than *in vivo*, so we have also made this amend throughout.

Reviewer comments to Author:

Reviewer: 1

Comments to the Author(s)

Amalina et al. have produced a detailed study of two compounds purported to be selective inhibitors on Bub1 kinase. This study is important because - as is well explained in the introduction - the role of Bub1,

specifically its kinase activity is contentious. Selective inhibitor(s) would help the field to get to the bottom of some of these discrepancies. What is presented here is a very carefully done study which confirms the in vitro inhibition of two compounds and tests the action of these compounds in cellulo, in part using a neat genetically encoded reporter of Bub1 activity. The results essentially underline that more development is required to obtain selective Bub1 kinase inhibitors.

The work is very high quality, and in my opinion scientifically sound. Note that I am not qualified to judge the chemistry aspects (synthesis of BAY-320). It is a useful contribution to a controversial area of cancer cell biology. I have some very minor comments that the authors can address straightforwardly.

We thank the reviewer for their detailed review, we have made all of the amends suggested as outlined below:

- Figure 7 the 10 μ M BAY-320 experiments need a time scale on the x-axis.
We have added the time scale to the axis
- Figure 7 the legend says 100 cells were analysed per condition. How are these selected? Random or some other method? In addition, if this is a single overnight imaging experiment (or a composite of different replicates) it should be stated in the legend.
We have amended the legend to state “One hundred randomly selected cells were analysed per condition in a single experiment”
- Figure 2 is the compression of 1.5-2 on the y-axis of the inhibition graph for 2OH-BNPP1 intentional?
Thank you, we have checked and the axis is correct – the compression was used to best use the available space.
- line 299 "to H2B to ectopically delocalised Bub1 kinase" -> "to H2B to ectopically delocalise Bub1 kinase"
Amend made as described
- Line 717 Ms. Excel -> Microsoft Excel
Amend made as described

Reviewer: 2

Comments to the Author(s)

There has been significant recent controversy in the literature regarding the role of Bub1 in mitosis. Conflicting results have been obtained using siRNA-mediated knockouts, CRISPR-mediated deletions, and small molecule-mediated inhibition. As there is significant clinical interest in Bub1 as a cancer target, it is important to develop an accurate picture of its cellular function. This new paper from the Taylor lab makes an excellent contribution toward that overall goal. Here, the authors carefully compare the effects of two putative Bub1 inhibitors in a series of in vitro and in vivo assays. They conclude that one inhibitor, BAY-320, is more potent against Bub1 in vivo, but they caution that it may exhibit some degree of promiscuous off-target binding. This is exactly the type of careful work that should be published.

I have two brief suggestions for improvement:

1) In certain locations, the authors make blanket statements about the effects of the 2OH drug, e.g., “BAY-320, but not 2OH-BNPP1, inhibits Bub1 in vivo”. The authors tested the 2OH drug at concentrations up to 10 μ M, but it is formally possible that they could see an inhibitory effect at 11 μ M, 20 μ M, or 200 μ M. Because of this consideration, I think that it is not appropriate to explicitly say “2OH-BNPP1 does not inhibit Bub1 in vivo”. The authors should temper this statement with modifiers - “does not effectively inhibit”, “does not inhibit at the concentrations tested”, etc., as they already do in several locations.

We agree with this suggestion and have amended as described, or to state the concentration, in various places throughout the manuscript. For example, the abstract (line 11), at the end of the introduction

(line 145), in the results (lines 222, 268, 282, 314, 322), legends to Figures 3 and 4, and in the discussion (line 398; 409).

2) The key experiment to demonstrate off-target effects of a mitotic kinase inhibitor is to treat cells carrying a genetic knockout of that drug's target with the drug itself (as done in for instance PMID: 29417930). If the authors hypothesize that 10uM treatment with BAY-320 has promiscuous off-target effects, then it would seem feasible to show that this concentration is also cytotoxic in the BUB1-deletion MEFs that the authors have generated. Please note: while I think that this experiment would strengthen the manuscript, if it is not feasible due to the time required for 4-OHT-mediated Bub1 disruption or for some other reason, I would still support publication without that experiment.

We agree with the reviewer that the experiment described would be required to definitively demonstrate any off-target effects of BAY-320. However, as also stated by the reviewer, this experiment would take considerable time, in particular due to the validation required to demonstrate that BAY-320 is acting as expected in the murine model used in the manuscript, and is therefore beyond the scope of this manuscript – especially as novel Bub1 inhibitors are now in development. Nonetheless, we feel this is an important point and we have therefore added it as a potential future experiment in the Discussion (line 475).

Reviewer: 3

Comments to the Author(s)

Amalina et al evaluate two small molecule inhibitors (2OH-BNPP1 and BAY-320) of the kinase Bub1 using in vitro and cell-based assays.

They find that in vitro both compounds inhibit Bub1 efficiently. However, in cells, only BAY-320 inhibits exogenous and endogenous Bub1 kinase activity. Because 4% remaining of Bub1 is sufficient for spindle checkpoint function, a concentration of BAY-320 is needed to fully inhibit Bub1 for functional analysis. At this concentration, the authors observe chromosome alignment defects, delayed mitotic progression, and cell death. Because cell death is mainly observed in interphase, the authors are concerned about potential off-target effects.

This study describes a series of well-designed and carefully carried out experiments clarifying the role of two Bub1 inhibitors. Small molecule kinase inhibitors are critical therapeutics in anti-cancer treatment and tools for biological investigations. Thus, careful characterization of their on- and off-target effects is essential, and this study makes an essential contribution to the evaluation of these compounds.

Minor points:

1. Why did the authors switch from 32P to Western blotting for the inhibitor testing? Please include an explanation.

The change was due to a lab move to a different building where use of γ -32P-ATP was not permitted. We agree that this should be clarified in the manuscript, therefore we have added a note in the methods (line 753).

2. The authors use two different kinase-dead mutations (D946N and K821R/K795R)), but only test K821R in vitro and D946N in cells. These results should be reconciled.

The reviewer raises a good point as the different effects of the different mutations has caused us (DP in particular) considerable anguish. In particular, why the DN complements the KO MEFs but the KR does not remains an unsolved mystery. However, with the different Bub1 assays, the kinase-dead mutants are there to serve as internal controls, not make conclusions about the underlying biology. So, in the *in vitro* assays, the point of the KR mutant is to confirm that the kinase activity being assayed is Bub1-dependent, rather than due to a co-purifying kinase. Similarly, in the HeLa cell assay, the DN mutant is to confirm that the ectopic phosphorylation of H2A is Bub1-dependent. We didn't test both mutants in both assays, we only tested the two in the MEF complementation assay.

3. Why did the authors pretreat DLD-1 cells for 3days with BAY-320 before evaluating mitotic duration? In the analysis of endogenous H2AT210 phosphorylation and Sgo1 localization, the authors only treated for 3 hrs? How was mitotic timing affected by shorter treatments? What was the mitotic index in cells treated for 3 days with BAY-320 compared to untreated or control-treated cells?

Apologies, the manuscript text was confusing here. Cells were actually treated then *immediately* imaged for 3 days, i.e. cells were not treated for 3 days prior to imaging. We have amended the text in the results to avoid confusion.

4. Why do the authors use various cell lines throughout the study? Please explain. It would be more informative if all experiments (analysis of chromosome alignment, SAC activity, mitotic timing, colony formation, and long-term survival) would be carried out consistently in at least one cell line, and others would be added as needed.

Various human cell lines were employed to optimally suit the experiment being undertaken. For example, HeLa and HEK293 cells were used for technical reasons as they have an established Flp-In™ T-Rex system for tetracycline-controlled expression of genes, and HEK-293 are the gold standard for recombinant protein expression. DLD-1 cells are ideal for timelapse microscopy as these cells remain ‘flatter’ during mitosis, meaning the chromosomes are more easily visible than cells of other lines that remain rounder (new text making this point on line 288).

MEFs were used for the SAC rescue experiments since we already had a conditional Bub1-knockout system established and fully characterised in these cells, as noted in the manuscript.

Finally, for the analysis of the impact of Bub1 inhibition on survival using colony assays, we utilised human cancer cell lines as Bub1 inhibitors are under development for their anti-cancer potential. RPE1 cells were used as a non-transformed comparator for these analyses, which were also used in the analysis of BAY-320 by Baron *et al.*, 2016. We agree that the reason for using the cancer cell lines should be noted in the manuscript and have therefore added text to the Results (line 310). Likewise RKO1 cells were utilised to evaluate the role of p53 in colony assays as they are wildtype for p53, allowing direct comparison of cells with and without p53 in an isogenic system - we have added text to clarify this in the Results (line 327).

However, note that we do already highlight in the Discussion that response to Bub1 inhibition may vary across different cell lines and an important next step is to evaluate the impact of Bub1 inhibition in additional cell lines. We also note in the discussion that the different cell lines used may have impacted our results.

===PREPARING YOUR MANUSCRIPT===

- one version should clearly identify all the changes that have been made (for instance, in coloured highlight, in bold text, or tracked changes);
- a 'clean' version of the new manuscript that incorporates the changes made, but does not highlight them. This version will be used for typesetting.

If you have been asked to revise the written English in your submission as a condition of publication, you must do so, and you are expected to provide evidence that you have received language editing support. The journal would prefer that you use a professional language editing service and provide a certificate of editing, but a signed letter from a colleague who is a proficient user of English is acceptable. Note the journal has arranged a number of discounts for authors using professional language editing services

(<https://royalsociety.org/journals/authors/benefits/language-editing/>).

===PREPARING YOUR REVISION IN SCHOLARONE===

-- Ensure that your data access statement meets the requirements

at <https://royalsociety.org/journals/authors/author-guidelines/#data>. You should ensure that you cite the dataset in your reference list. If you have deposited data etc in the Dryad repository, please only include the 'For publication' link at this stage. You should remove the 'For review' link.

-- If you are requesting an article processing charge waiver, you must select the relevant waiver option (if requesting a discretionary waiver, the form should have been uploaded, see 'File upload' above).

-- If you have uploaded any electronic supplementary (ESM) files, please ensure you follow the guidance at <https://royalsociety.org/journals/authors/author-guidelines/#supplementary-material> to include a suitable title and informative caption. An example of appropriate titling and captioning may be found at https://figshare.com/articles/Table_S2_from_Is_there_a_trade-off_between_peak_performance_and_performance_breadth_across_temperatures_for_aerobic_scope_in_teleost_fishes_/3843624.

Journal Name: Royal Society Open Science

Journal Code: RSOS

Online ISSN: 2054-5703

Journal Admin Email: openscience@royalsociety.org

Journal Editor: Andrew Dunn

Journal Editor Email: openscience@royalsociety.org

MS Reference Number: RSOS-210854

Article Status: SUBMITTED

MS Dryad ID: RSOS-210854

MS Title: Inhibitors of the Bub1 spindle assembly checkpoint kinase: Synthesis of BAY-320 and comparison with 2OH-BNPP1

MS Authors: Amalina, Ilma; Bennett, Ailsa; Whalley, Helen; Perera, David; McGrail, Joanne; Tighe, Anthony; Procter, David; Taylor, Stephen

Contact Author: Stephen Taylor

Contact Author Email: stephen.taylor@manchester.ac.uk

Contact Author Address 1: Wilmslow Road

Contact Author Address 2:

Contact Author Address 3:

Contact Author City: Manchester

Contact Author State:

Contact Author Country: United Kingdom of Great Britain and Northern Ireland

Contact Author ZIP/Postal Code: M20 4QL

Keywords: Bub1, Spindle Assembly Checkpoint, BAY-320, 2OH-BNPP1, Kinase Inhibitor

Abstract: Bub1 is a serine/threonine kinase proposed to function centrally in both mitotic chromosome alignment and the spindle assembly checkpoint (SAC), however its role remains controversial. Although it is well documented that Bub1 phosphorylation of Histone 2A at T120 (H2ApT120) recruits Sgo1/2 to kinetochores, the requirement of its kinase activity for chromosome alignment and the SAC is debated. As small-molecule inhibitors can be invaluable tools for investigation of kinase function, we decided to evaluate the relative potential of two agents (2OH-BNPP1 and BAY-320) as Bub1 inhibitors. After confirming that both agents inhibit Bub1 in vitro, we developed a cell based-assay to specifically measure Bub1 inhibition in vivo. For this assay we overexpressed a fusion of Histone 2B and the Bub1 kinase region (Bub1C) tethering it in close proximity to H2A, which generated a strong ectopic H2ApT120 signal along chromosome arms. The ectopic signal generated from Bub1C activity was effectively inhibited by BAY-320, but not 2OH-BNPP1. In addition, only BAY-320 was able to inhibit endogenous Bub1-mediated Sgo1 localisation. Preliminary experiments using BAY-320 suggested a minor role for Bub1 kinase activity in chromosome alignment and the SAC, however results suggest that BAY-320 may exhibit off-target effects at the concentration required to demonstrate these outcomes. In conclusion, 2OH-BNPP1 may not be an effective Bub1 inhibitor in vivo, and while BAY-320 is able to inhibit Bub1 in vivo, the high concentrations required and potential for off-target effects highlight the ongoing need for improved Bub1 inhibitors.

EndDryadContent